# Blur to Focus Attention in Fine-Grained Visual Recognition

## Abstract

Fine-grained visual recognition (FGVR) requires distinguishing categories separated by tiny discriminative cues such as fine textures, part shapes, or color patterns. In typical datasets, discriminative regions occupy less than 30% of the image area, and in ultra-fine-grained cases often under 10%. This sparsity makes training highly fragile. Standard data augmentations risk destroying these subtle signals, while part-based or attention-driven models depend on annotations or rigid architectures and often fail under pose variation, occlusion, or cluttered backgrounds. We present DEFOCA, a simple layer that patchifies an image and stochastically applies Gaussian blur to selected patches. Each patch selection (*e.g.*, random or contiguous) defines a single view, and multiple such views encourage the model to rely on diverse subsets of discriminative cues while reducing dependence on spurious background features. In this way, DEFOCA functions as a soft, attention-like mechanism that integrates seamlessly with existing architectures. Theoretically, we show that DEFOCA is label-safe, that contiguous patch layouts maximize the probability of label-safety, and that the expected representation drift is minimized. This guarantees that critical features are preserved while irrelevant high-frequency noise is suppressed, thereby narrowing the generalization gap. Empirically, DEFOCA achieves competitive performance on widely used fine-grained benchmarks (CUB-200-2011, Stanford Cars, NABirds, FGVC Aircraft) as well as ultra-fine-grained datasets (Cotton80, SoyGene, SoyGlobal). These results establish DEFOCA as a principled and highly effective solution for robust and discriminative feature learning in FGVR.

## 1 Introduction

Fine-grained visual recognition (FGVR) requires distinguishing categories that differ in subtle, localized cues, such as bird species, car models, or plant and leaf types. Unlike coarse-grained recognition, which often rely on global shapes or dominant colors, FGVR depends on capturing minute textures, part structures, and fine-grained patterns. In typical datasets, discriminative regions occupy less than 30% of the image area, and in ultra-fine-grained cases often under 10% (He et al., 2019). In real-world settings, these discriminative cues are easily obscured by occlusion, changes in viewpoint, pose variations, or cluttered backgrounds. Standard augmentations or transformation strategies developed for large-scale recognition datasets (*e.g.*, ImageNet (Deng et al., 2009)) can further distort or erase these critical details, weakening the subtle distinctions that FGVR relies on.

Existing solutions address these challenges using part-based models or attention mechanisms that explicitly localize discriminative regions. While effective under controlled conditions, these methods are highly model- and dataset-dependent: they often require expert-level, high-quality part annotations, rely on pre-trained detectors, or involve costly architectural modifications(Chu et al., 2021). Their performance degrades when discriminative cues are partially visible, occluded, or spatially diffuse. Likewise, conventional augmentation strategies, *e.g.*, random crops, CutMix(Yun et al., 2019), or DropBlock(Ghiasi et al., 2018), ignore spatial structure of critical features and can inadvertently erase the very details FGVR depends on, limiting generalization to real-world conditions.

Inspired by masked image modeling (MIM)(He et al., 2022b; Bao et al., 2022), which encourages networks to infer missing information by masking patches, we introduce DEFOCA that applies Gaussian blur to a set of randomly selected image patches (Fig. 1, left). Unlike MIM, which removes

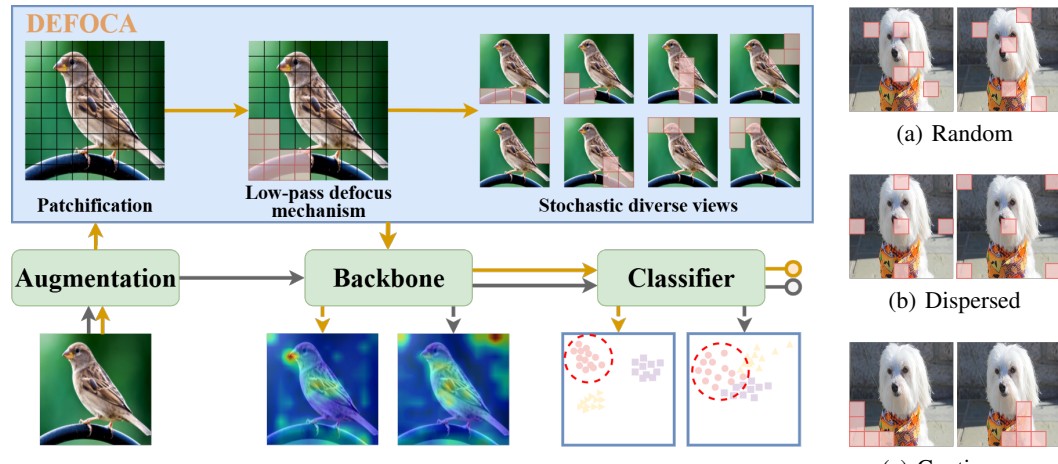

Figure 1: (**Left**) Comparison between a standard recognition pipeline (black arrows) and the same pipeline with the DEFOCA layer integrated (orange arrows) for fine- and ultra-fine-grained visual recognition. After global augmentation, DEFOCA applies (i) patchification, (ii) a low-pass defocus operation, and (iii) stochastic generation of multiple diverse views. Integrating DEFOCA produces sharper, more distinctive attention maps, realizing our *blur-to-focus attention mechanism* and yielding more separable, compact feature clusters for improved discriminability (dashed arrows). (**Right**) Patch selection strategies in DEFOCA. (a): *Random* places blurred regions arbitrarily, often lacking spatial structure; (b): *Dispersed* enforces separation between patches, increasing coverage but fragmenting context; (c): *Contiguous* selects coherent regions, preserving local semantics and context.

patches entirely for self-supervised reconstruction, DEFOCA gradually blurs selected regions, preserving sufficient spatial structure for class-relevant information. By blurring rather than masking, it suppresses high-frequency noise while maintaining both local details and global context, achieving a balance between fine-grained cues and overall structure. Although the blurred regions are not deterministically "non-critical", this random degradation encourages the network to focus on the remaining informative regions, naturally guiding attention toward discriminative cues. We explore different patch selection strategies, including random, dispersed, and contiguous layouts (Fig. 1, right). Using contiguous regions encourages the network to rely on coherent, semantically meaningful areas rather than fragmented signals, improving feature stability and robustness. Applying multiple stochastic variations of blurred patches per input generates diverse views, which act as a soft attention mechanism, naturally regularizing the network and improving its robustness to occlusion, pose variation, and background clutter. Unlike attention- or part-based approaches, DEFOCA requires no annotations or architectural redesign, and unlike conventional augmentations or feature transformations, it preserves discriminative cues while reducing reliance on irrelevant or noisy information.

Beyond its practical design, DEFOCA is supported by a strong theoretical rationale. Using contiguous patches maximizes label-safety, bounds representation drift, and helps preserve discriminative features, while stochastic multi-view applications naturally regularize the network, reducing overfitting and narrowing the generalization gap. This principled combination of selective degradation and multi-view training enhances both feature discriminability and robustness, making DEFOCA highly effective for FGVR tasks. We evaluate DEFOCA on standard FGVR datasets (*e.g.*, CUB-200-2011, Stanford Cars, NABirds, FGVC Aircraft) as well as ultra-fine-grained benchmarks (*e.g.*, Cotton80, SoyLocal, SoyGene). Across all datasets, DEFOCA consistently improves performance and achieves competitive results, demonstrating its simplicity, versatility, and effectiveness in learning robust, discriminative representations. Our main **contributions** are as follows:

i. We introduce DEFOCA, a stochastic defocus layer that encourages networks to focus on discriminative regions while generating diverse multi-view variants. It provides natural and effective regularization, functioning as a *parameter-free attention mechanism*.

ii. We present a theoretically grounded formulation, showing that DEFOCA preserves label-critical patches, bounds representation drift, and selectively suppresses irrelevant high-frequency features, thereby narrowing the generalization gap.

iii. We evaluate DEFOCA on both fine-grained and ultra-fine-grained benchmarks, achieving competitive performance without requiring additional annotations or architectural modifications.

## 2 APPROACH

In Appendix A.1, we review closely related works on FGVR, including data augmentation and regularization techniques, attention- and part-based mechanisms, and existing theoretical foundations. We also highlight how our approach differs from prior work. Below, we introduce our DEFOCA.

### 2.1 DEFOCA: A BLUR-TO-FOCUS ATTENTION LAYER

**Notation.** We consider the FGVR setting, where each input contains subtle, localized cues that are critical for distinguishing classes. Let $x \in \mathbb{X}$ denote an input image tensor. We partition $x$ into $N = P \times P$ non-overlapping patches: $x = \{p_1, p_2, \ldots, p_N\}$, where $p_i$ denotes the $i$-th patch and $P$ is the number of patches along each spatial dimension. Among the $N$ patches, we define an index set $S \subseteq \{1, \ldots, N\}$ of size $s = |S| \ll N$, corresponding to the discriminative patches $\{p_i : i \in S\}$. These patches capture fine textures, part shapes, or localized color patterns that are essential for correct classification. We denote a transformation that modifies a subset of patches as a function $\mathcal{T} : \mathbb{X} \to \mathbb{X}$. Applying $\mathcal{T}$ to $x$ produces a transformed tensor $x' = \mathcal{T}(x)$, where the specific patches to modify may be chosen stochastically.

**Patch-based low-pass mechanism.** In DEFOCA, the transformation $\mathcal{T}$ is realized as a stochastic, patch-based low-pass operation. Specifically, $\mathcal{T}$ selects a subset of $n$ patches and applies a low-pass filter to them, while leaving the remaining patches unchanged. For Gaussian blur, the filtering strength is controlled by the standard deviation $\sigma$, which determines the degree of smoothing. By selectively blurring patches, the network is encouraged to rely on the remaining unblurred regions that carry class-relevant information, such as fine textures, part shapes, or color patterns.

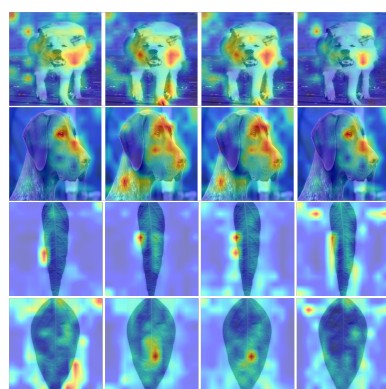

Figure 2: Attention maps for different patch selection strategies. From left to right: baseline (no DEFOCA), random, contiguous, and dispersed. Random and contiguous strategies capture clearer, more discriminative cues (*e.g.*, dog's cheek, facial contours, leaf veins), while baseline and dispersed attention maps appear similar and less focused.

Formally, Gaussian blurring is naturally described in the Fourier domain as attenuation of high-frequency components. For a selected patch $p_i$, let $\mathcal{F}(p_i)(\omega)$ denote its Fourier transform at spatial frequency $\omega$, where $\omega$ represents the coordinates in the Fourier domain. The filtered patch is:

$$p'_i = \mathcal{F}^{-1}(H_{\sigma_i}(\omega) \cdot \mathcal{F}(p_i)(\omega)), \tag{1}$$

where $H_{\sigma_i}(\omega) = \exp\left(-\frac{\|\omega\|^2}{2\sigma_i^2}\right)$ is the Gaussian transfer function parameterized by the standard deviation $\sigma_i$. This formulation highlights that low-frequency components ($\|\omega\|$ small) pass through largely unchanged, while high-frequency components ($\|\omega\|$ large) are increasingly suppressed as $\sigma_i$ grows. Consequently, the operation systematically reduces rapid pixel-to-pixel variations (fine textures or noise) while preserving coarse, class-relevant structures. Non-selected patches remain intact, and the filtered and unfiltered patches are recombined to yield the transformed input $x'$.

**Blur-to-focus attention.** Patch selection in DEFOCA is stochastic, generating different combinations of patches across training iterations to ensure broad coverage of the input and to produce multiple diverse views per input (denoted as $V$, We set $V = 8$ for simplicity). We compare different patch selection strategies, including random, dispersed, and contiguous (see Fig. 1, right), and find that random and contiguous strategies capture clearer, more discriminative cues, focusing attention on meaningful regions such as the dog's cheek, facial contours, and leaf veins and textures (see Fig. 2). We also explore alternative patch operations beyond Gaussian blur (low-pass), including high-pass, pepper noise, and color jitter. As shown in Fig. 3, low-pass produces the clearest and most focused attention maps compared to the other operations. When trained with low-pass DEFOCA, the network focuses more on key regions critical for FGVR, including the bird's beak, paw, feathers, and

body shape. These analyses motivate the use of the *contiguous low-pass mechanism* in DEFOCA, which we refer to as the *blur-to-focus attention* layer.

**DEFOCA applied in training, removed in testing.** DEFOCA is applied on-the-fly during training to generate multiple stochastic views of each input without modifying the network architecture. It operates directly on input images and is applied after common global augmentations such as random cropping, flipping, or color jitter. This ensures full compatibility with standard pipelines, allowing the network to benefit from both global transformations and localized stochastic degradations. At test time, DEFOCA is removed, as the network has already learned to attend to discriminative regions, eliminating the need for additional operations while maintaining improved feature focus.

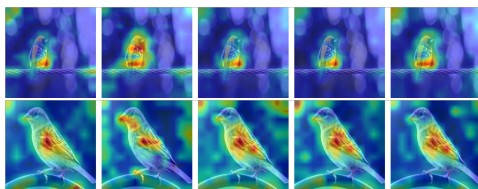

Figure 3: Attention maps for different patch operations. From left to right: baseline, low-pass (ours), high-pass, pepper noise, and color jitter. Low-pass in DEFOCA produces the clearest focus on key regions, *e.g.*, the bird's beak, paw, feathers, and body shape.

**Practical considerations.** The main hyperparameter of DEFOCA is the number of patches per spatial dimension, $P$, which controls the granularity of the patch grid. Smaller patches (larger $P$) enable finer, more localized suppression, highlighting subtle discriminative cues, whereas larger patches (smaller $P$) preserve broader contextual information. The Gaussian blur standard deviation $\sigma$ adjusts the filtering strength for each patch. Contiguous patch selection ensures spatial coherence, producing smooth, natural-looking patterns that avoid fragmented artifacts. The patch selection ratio $(n/N)$ controls how many patches are blurred, balancing the degree of localized suppression against the preservation of global context. By balancing patch size, filter strength, and patch selection ratio, DEFOCA encourages the network to attend to both fine-grained local features and relevant global context, resulting in robust, discriminative representations that generalize well across pose variations, occlusions, and cluttered backgrounds. Below, we provide a conceptual and mathematical perspective on how DEFOCA implicitly guides the network's attention toward class-relevant regions.

## 2.2 CONNECTION TO ATTENTION MECHANISMS

Although DEFOCA does not compute explicit attention weights, its stochastic patch-based low-pass operation can be interpreted as a form of implicit, soft attention. Let $\boldsymbol{x} = \{\boldsymbol{p}_1, \ldots, \boldsymbol{p}_N\}$ denote the ordered collection of patches defined in the previous section, and let $\tilde{\boldsymbol{p}}_i$ be the low-pass filtered version of patch $\boldsymbol{p}_i$. DEFOCA can be described in two complementary ways. First, in a *soft* formulation, each patch is a weighted combination of its original and filtered versions:

$$\boldsymbol{p}_i' = \alpha_i \tilde{\boldsymbol{p}}_i + (1 - \alpha_i)\boldsymbol{p}_i, \quad 0 \leq \alpha_i < 1, \tag{2}$$

where $\alpha_i$ controls the strength of high-frequency suppression.[1] Smaller $\alpha_i$ preserves more of the original patch content, analogous to assigning higher attention to discriminative regions, whereas larger $\alpha_i$ increases attenuation, reducing the patch's influence. Second, in a *hard* stochastic formulation, a binary mask $\boldsymbol{m} = (m_1, \ldots, m_N)$ indicates which patches are blurred:

$$\boldsymbol{p}_i' = m_i \boldsymbol{p}_i + (1 - m_i)\tilde{\boldsymbol{p}}_i, \tag{3}$$

where $m_i = 1$ if the patch is left unblurred and $0$ if it is blurred. Over multiple stochastic applications, each patch is sometimes blurred and sometimes left intact, generating a distribution of *attention masks*. Taking the expectation over the stochastic mask distribution $\mathcal{P}$,

$$\mathbb{E}_{\boldsymbol{m} \sim \mathcal{P}}[\boldsymbol{x}'] = \mathbb{E}_{\boldsymbol{m} \sim \mathcal{P}}\left[\sum_{i=1}^{N} m_i \boldsymbol{p}_i + (1 - m_i)\tilde{\boldsymbol{p}}_i\right], \tag{4}$$

makes explicit that, on average, the network places greater emphasis on the unblurred, informative patches while attenuating the contribution of blurred or low-pass filtered regions. This formalizes the notion that DEFOCA implicitly implements a soft, stochastic attention mechanism over the input.

---

[1]Here, $\alpha_i$ is used as a conceptual interpolation to illustrate soft attention; in our implementation, we use hard masking (binary $m_i$) for stochastic patch selection.

This framework provides several **key insights**. First, DEFOCA acts as a stochastic attention mechanism, guiding the network to focus on discriminative regions without requiring explicit attention modules or part annotations. Second, the patch granularity $P$ and filtering strength $\sigma$ offer a principled trade-off between local and global information: smaller patches concentrate on subtle, fine-grained cues, while larger patches preserve broader contextual structures. Finally, the stochastic, multi-view application of DEFOCA introduces implicit regularization, improving robustness to occlusion, pose variation, and background clutter, while stabilizing the learned representations.

## 2.3 THEORETICAL ANALYSIS OF DEFOCA

We provide a theoretical analysis of DEFOCA, illustrating how its selective, stochastic patch-based low-pass operation improves robustness while preserving discriminative features in fine-grained visual recognition. Let $f_{\boldsymbol{\theta}} : \mathbb{X} \to \mathbb{R}^d$ denote the feature representation extracted by a neural network parameterized by $\boldsymbol{\theta}$. We assume that $f_{\boldsymbol{\theta}}$ is $L$-Lipschitz with respect to patch-wise modifications, meaning that for any patch-altering transformation $\mathcal{T}$,

$$\|f_{\boldsymbol{\theta}}(\boldsymbol{x}) - f_{\boldsymbol{\theta}}(\mathcal{T}(\boldsymbol{x}))\|_2 \le L \sum_{i=1}^{N} \mathbf{1}_{\{i \in \mathcal{M}(\mathcal{T})\}}, \tag{5}$$

where $\mathcal{M}(\mathcal{T})$ denotes the set of patches modified by $\mathcal{T}$. This condition formalizes the intuition that modifications to non-discriminative patches induce limited changes in the feature representation, whereas altering discriminative patches can produce substantial deviations. Within this framework, the goal is to design transformations that selectively suppress irrelevant high-frequency components in non-critical regions while preserving the discriminative patches, ensuring that the transformed input $\boldsymbol{x}' = \mathcal{T}(\boldsymbol{x})$ remains label-safe, *i.e.*, the class label is preserved. This formulation provides a principled foundation for analyzing DEFOCA's theoretical properties, including label-safety probability, representation drift, signal-to-noise ratio enhancement, and generalization guarantees.

**Label-safety probability.** In fine-grained recognition, it is crucial that transformations do not destroy the discriminative information required for correct classification. We formalize this requirement through the concept of label-safety.

**Definition 1** (Label-safe transformation). *A transformation $\mathcal{T}$ is said to be label-safe if it modifies only non-discriminative patches,* i.e.*,*

$$\mathcal{T}(\boldsymbol{p}_i) = \boldsymbol{p}_i \quad \forall i \in S, \tag{6}$$

*where $S$ denotes the set of discriminative patches. In other words, label-safety guarantees that the transformed image $\boldsymbol{x}' = \mathcal{T}(\boldsymbol{x})$ preserves all critical information necessary for correct classification.*

We now quantify the probability that a randomly chosen transformation is label-safe. Suppose $\mathcal{T}$ modifies exactly $n$ patches out of $N$, selected uniformly at random. The total number of possible sets of $n$ patches is $\binom{N}{n}$. Among these, the number of sets that avoid all discriminative patches is $\binom{N-s}{n}$, where $s = |S|$ is the number of discriminative patches. Therefore, the combinatorial probability that a random transformation is label-safe is $P_{\text{safe}} = \frac{\binom{N-s}{n}}{\binom{N}{n}}$.

This probability provides a concrete measure of how likely a transformation preserves critical features. In fine-grained recognition scenarios, $s \ll N$, meaning that discriminative patches are sparse but crucial (He et al., 2019). Consequently, naïve random modifications often alter one or more discriminative patches, resulting in label corruption. In contrast, structured transformations that apply contiguous or guided modifications, as in DEFOCA, can increase $P_{\text{safe}}$, ensuring that transformed images remain consistent with their labels. This combinatorial perspective provides the foundation for analyzing representation drift and the generalization benefits of label-safe transformations.

**Representation drift.** Having formalized label-safety, we now study its impact on the stability of model representations under patch-based transformations. Intuitively, modifying non-discriminative patches should have minimal effect on the feature embedding, whereas altering discriminative patches can cause substantial deviations. We formalize this notion as representation drift.

**Lemma 1** (Expected representation drift). *Let $\boldsymbol{x}' = \mathcal{T}(\boldsymbol{x})$ be a transformed input obtained by applying the stochastic patch-based transformation $\mathcal{T}$ to $\boldsymbol{x}$. The expected squared drift of the representation is bounded as*

$$\mathbb{E}_{\boldsymbol{x}' \sim \mathcal{T}(\boldsymbol{x})} \left[ \|f_{\boldsymbol{\theta}}(\boldsymbol{x}) - f_{\boldsymbol{\theta}}(\boldsymbol{x}')\|_2^2 \right] \le P_{\text{safe}} \, (Ln)^2 + (1 - P_{\text{safe}}) \, M^2, \tag{7}$$

*where $L$ is the Lipschitz constant of $f_{\boldsymbol{\theta}}$ with respect to patch modifications, $n$ is the number of modified patches, $P_{safe}$ is the probability that $\mathcal{T}$ does not alter any discriminative patch, and $M \gg Ln$ is an upper bound on the drift if at least one discriminative patch is modified.*

See Appendix A.2 for the proof. This lemma establishes a clear connection between label-safety and robust feature representations: by maximizing $P_{\text{safe}}$, for example through contiguous patch selection as in DEFOCA, we ensure that the network's representations remain stable and discriminative, providing theoretical justification for improved robustness in FGVR tasks.

**High-frequency noise suppression.** Building on the patch representation introduced earlier, DE-FOCA selectively attenuates high-frequency noise in non-discriminative patches, while preserving discriminative patches. For $i \notin S$, the filtered patch in the Fourier domain is $\hat{\boldsymbol{p}}'_i(\omega) = H_{\sigma_i}(\omega) \cdot \mathcal{F}(\boldsymbol{p}_i)(\omega)$, while for $i \in S$, the patch remains unchanged: $\hat{\boldsymbol{p}}'_i(\omega) = \mathcal{F}(\boldsymbol{p}_i)(\omega)$. This operation systematically reduces irrelevant high-frequency components that could distract the network while preserving the essential low-frequency structures that encode class-critical information.

**Proposition 1** (Signal-to-noise ratio enhancement). *Selective low-pass filtering strictly increases the signal-to-noise ratio (SNR) of discriminative patches. Formally, define*

$$SNR = \frac{\sum_{i \in S} \|\mathcal{F}(\boldsymbol{p}_i)\|^2}{\sum_{i \notin S} \|\mathcal{F}(\boldsymbol{p}_i)\|^2}, \qquad SNR' = \frac{\sum_{i \in S} \|\mathcal{F}(\boldsymbol{p}_i)\|^2}{\sum_{i \notin S} \|H_{\sigma_i}(\omega) \cdot \mathcal{F}(\boldsymbol{p}_i)\|^2}. \tag{8}$$

*Since $0 \le H_{\sigma_i}(\omega) \le 1$ and $H_{\sigma_i}(\omega) < 1$ for high frequencies, we have $SNR' > SNR$.*

The proof can be found in Appendix A.3. This analysis demonstrates that DEFOCA simultaneously preserves essential discriminative features and reduces noisy, potentially misleading content. By improving the SNR of key patches, the network is implicitly guided to focus on informative regions, complementing the label-safety and representation drift properties established earlier.

**Expected loss under transformation.** Let $\ell(f_{\boldsymbol{\theta}}(\boldsymbol{x}), y)$ denote the loss of predicting label $y$ from input $\boldsymbol{x}$. Consider a patch-based transformation $\mathcal{T}$ producing $\boldsymbol{x}' = \mathcal{T}(\boldsymbol{x})$. By Lipschitz continuity of $f_{\boldsymbol{\theta}}$ and the definition of representation drift, we have

$$\mathcal{L}_{\text{DEFOCA}} := \mathbb{E}_{\boldsymbol{x}' \sim \mathcal{T}(\boldsymbol{x})}[\ell(f_{\boldsymbol{\theta}}(\boldsymbol{x}'), y)] \le \ell(f_{\boldsymbol{\theta}}(\boldsymbol{x}), y) + \mathbb{E}_{\boldsymbol{x}' \sim \mathcal{T}(\boldsymbol{x})}[\|f_{\boldsymbol{\theta}}(\boldsymbol{x}) - f_{\boldsymbol{\theta}}(\boldsymbol{x}')\|_2^2]. \tag{9}$$

Substituting the expected representation drift from Lemma 1, we obtain

$$\mathcal{L}_{\text{DEFOCA}} \le \ell(f_{\boldsymbol{\theta}}(\boldsymbol{x}), y) + P_{\text{safe}}(Ln)^2 + (1 - P_{\text{safe}})M^2, \tag{10}$$

where $P_{\text{safe}}$ is the label-safety probability, $Ln$ bounds the drift for label-safe transformations, and $M \gg Ln$ bounds the drift when at least one discriminative patch is modified.

This formalizes the intuition that transformations which maximize $P_{\text{safe}}$, such as DEFOCA's contiguous patch blurring, preserve critical features while minimally increasing the expected loss. Furthermore, selective low-pass filtering attenuates non-critical high-frequency components, enhancing the stability of the learned representation. These effects show that our patch-based transformations can improve robustness and generalization without compromising label fidelity.

**Generalization guarantee.** DEFOCA not only stabilizes representations but also increases effective dataset diversity while preserving label-critical patches. To formalize this, we use a PAC-Bayes style generalization bound. Let $m$ denote the number of training samples, $\hat{R}_{\text{DEFOCA}}(f_{\boldsymbol{\theta}})$ the empirical risk under DEFOCA-transformed samples, and $R(f_{\boldsymbol{\theta}})$ the true risk. For a posterior distribution $Q$ over parameters and a prior $P$, with confidence $1 - \delta$, the PAC-Bayes theorem gives

$$R(f_{\boldsymbol{\theta}}) \le \hat{R}_{\text{DEFOCA}}(f_{\boldsymbol{\theta}}) + \frac{1}{2m}\left[D_{\text{KL}}(Q\|P) + \log\frac{1}{\delta}\right]. \tag{11}$$

Here, $Q$ can be any posterior distribution over the network parameters, highlighting the generality of the bound. Crucially, the empirical risk under transformation can be bounded using the expected loss derived in the previous section:

$$\hat{R}_{\text{DEFOCA}}(f_{\boldsymbol{\theta}}) = \frac{1}{m}\sum_{i=1}^{m}\mathcal{L}_{\text{DEFOCA}}(\boldsymbol{x}_i, y_i) \le \frac{1}{m}\sum_{i=1}^{m}\left[\ell(f_{\boldsymbol{\theta}}(\boldsymbol{x}_i), y_i) + P_{\text{safe}}(Ln)^2 + (1 - P_{\text{safe}})M^2\right].$$

$$\tag{12}$$

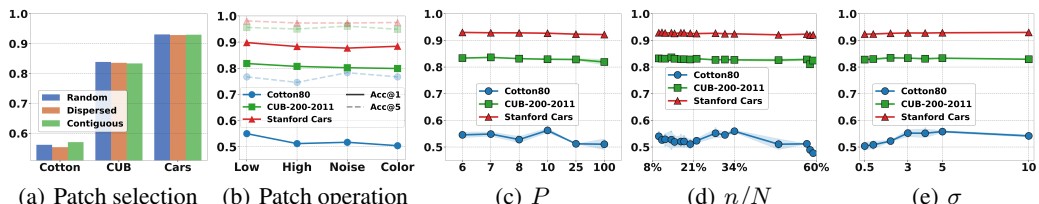

| (a) Patch selection | (b) Patch operation | (c) $P$ | (d) $n/N$ | (e) $\sigma$ |

Figure 4: Variant studies on (a) patch selection strategies and (b) patch operations, including low-pass, high-pass, pepper noise, and color jitter. Hyperparameter evaluations for (c) grid size $P$, (d) patch selection ratio $n/N$, and (e) Gaussian blur strength $\sigma$.

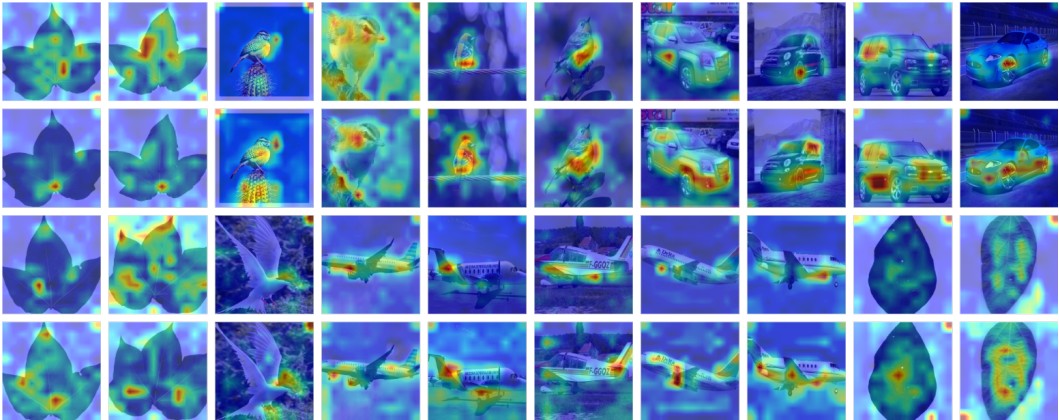

Figure 5: Attention map comparisons on five benchmarks. Rows 1 and 3: baseline (Tiny ViT); Rows 2 and 4: DEFOCA. Last-layer attention overlaid (warmer colors = higher attention). DEFOCA produces sharper, more localized focus on discriminative parts (*e.g.*, bird heads/wings, car grilles/lights, aircraft engines/tails, leaf veins), while baselines spread attention diffusely or onto the background.

Since $P_{\text{safe}}$ is maximized by DEFOCA's structured or contiguous patch modifications, most transformations are label-safe, and the additive term $(1-P_{\text{safe}})M^2$ is minimized. This effectively increases sample diversity without compromising labels, leading to a smaller empirical risk and a provable reduction of generalization gap. Combined with representation drift and high-frequency suppression analyses, this establishes a rigorous theoretical guarantee that DEFOCA improves FGVR robustness.

## 3 EXPERIMENT

### 3.1 SETUP

**Datasets.** We evaluate DEFOCA on multiple benchmarks. Experiments are conducted on widely used fine-grained datasets: CUB-200-2011(Wah et al., 2011), Stanford Cars(Krause et al., 2013), NABirds(Van Horn et al., 2015), and FGVC-Aircraft(Maji et al., 2013), and ultra-fine-grained datasets(Yu et al., 2021b), including Cotton80, SoyLocal, SoyGlobal, SoyAgeing, and SoyGene. Dataset details and statistics are in Appendix A.4.

**Setups.** DEFOCA is integrated into popular backbones, *e.g.*, ResNet34, ResNet50, and vision transformers (ViT), without altering their architectures. Images are resized to $224 \times 224$ and undergo standard augmentations, after which DEFOCA is applied. CNNs are trained with SGD (momentum 0.9, learning rate 0.01) and transformers with AdamW (learning rate $3 \times 10^{-4}$) for 160 epochs, using batch sizes of 16. We explore $P \in \{6, 7, 8, 10, 25, 100\}$, $\sigma \in \{0.5, 1, 2, 3, 4, 5, 10\}$, and patch selection ratios $n/N$ between 10% and 70%. DEFOCA is applied on-the-fly with negligible computational overhead. Detailed experimental setups are provided in the Appendix.

**Baselines & protocols.** We compare DEFOCA against several representative models, including: (i) standard backbones such as ResNet34, ResNet50, and ViT (21M parameters); (ii) attention-based models that explicitly highlight discriminative regions via spatial or channel-wise mechanisms; and (iii) existing state-of-the-art approaches. Models are evaluated using top-1 accuracy and total train-

Table 1: Comparison on FGVR benchmarks. All values are in %. DEFOCA-Tiny ViT consistently achieves competitive performance, demonstrating its ability to capture subtle fine-grained distinctions. Notably, our model achieves strong performance with just 21.20M parameters, while the top performer exceeds it by 1.6% using 5.3× more parameters (112.42M *vs.* 21.20M).

| Method | Aircraft | CUB-200 | Stanford Cars | NABirds | Mean | #Params(M) |
|---|---|---|---|---|---|---|
| ResNet34 (He et al., 2016) | 84.7 | 77.0 | 90.6 | 80.2 | 83.1 | 21.80 |
| ResNet50 (He et al., 2016) | 91.5 | 84.5 | 91.8 | 83.5 | 87.8 | 25.56 |
| Tiny ViT (Wu et al., 2022) | 90.7 | 88.6 | 93.5 | 87.5 | 90.0 | 21.20 |
| NTS-Net (Yang et al., 2018) | 91.4 | 87.5 | 93.9 | N/A | 90.9 | 50.20 |
| Horospherical-R34(Berg et al., 2024) | N/A | 59.2 | 82.9 | N/A | 70.5 | 18.20 |
| ISDA-R50 (Wang et al., 2019) | 91.7 | 85.3 | 93.2 | 83.9 | 88.5 | 25.56 |
| LSDA-R50 (Pu et al., 2024) | 92.7 | 86.7 | 94.3 | 85.5 | 89.8 | 33.86 |
| ACNet-R50 (Ji et al., 2020) | 92.4 | 88.1 | 94.6 | N/A | 91.7 | 25.56 |
| Cross-X-R50 (Luo et al., 2019) | 92.6 | 87.7 | 94.6 | 86.2 | 90.2 | 25.56 |
| MaxEnt-CNN (Dubey et al., 2018b) | 89.8 | 80.4 | 93.9 | 83.0 | 86.7 | 28.68 |
| FFTV-ViT (Wang et al., 2021) | N/A | 91.4 | N/A | 89.5 | 90.4 | 86.86 |
| TransFG-ViT (He et al., 2022a) | N/A | 91.3 | 94.8 | 90.2 | 92.1 | 86.86 |
| ProtoPool (Rymarczyk et al., 2022) | N/A | 87.6 | 91.6 | N/A | 89.6 | 25.56 |
| DCAL-R50+ViT (Zhu et al., 2022) | **93.3** | **92.0** | **95.3** | N/A | **93.5** | 112.42 |
| PEDTans-ViT (Lin et al., 2022) | N/A | 91.7 | 95.1 | 90.7 | 92.5 | 86.86 |
| TransIFC-ViT (Liu et al., 2023) | N/A | 91.0 | N/A | **90.9** | 90.9 | 196.74 |
| INT-NFR50 (Xiong et al., 2025) | 64.9 | 62.6 | N/A | N/A | 63.7 | 25.56 |
| FIDO (Korsch et al., 2025) | N/A | 90.9 | N/A | 89.3 | 90.1 | 25.56 |
| UniFGVC (Guo et al., 2025) | 61.1 | 78.8 | 94.6 | N/A | 78.1 | 117.11 |
| DEFOCA-ResNet34 | 86.0 | 78.7 | 92.2 | 81.0 | 84.4 | 21.80 |
| DEFOCA-ResNet50 | 91.5 | 85.8 | 92.9 | 84.9 | 88.7 | 25.56 |
| DEFOCA-Tiny ViT | 92.8 | 90.7 | 94.6 | 89.8 | 91.9 | **21.20** |

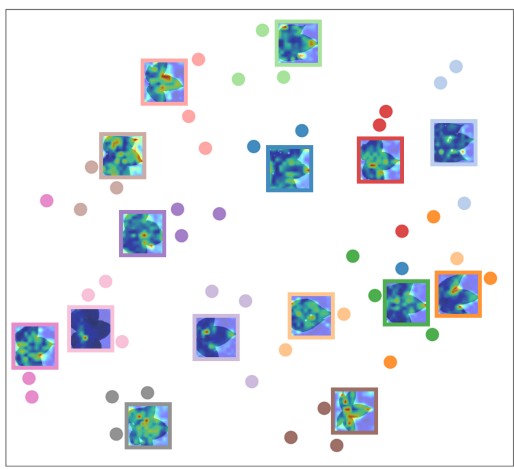 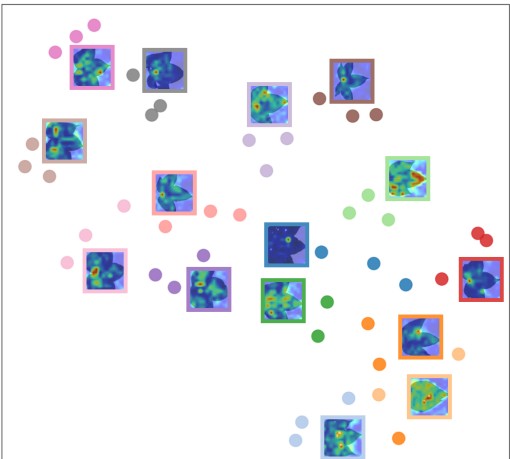

(a) Baseline (no DEFOCA)        (b) With DEFOCA

Figure 6: t-SNE of Cotton80 test set: (a) baseline (Tiny ViT), (b) with DEFOCA. Thumbnails show one sample per class; bounding boxes and dot colors indicate class. DEFOCA produces tighter clusters and more focused attention on discriminative regions.

able parameters. Attention maps and t-SNE visualizations illustrate the learned representations, highlighting feature separability and structure.

## 3.2 EVALUATION

Qualitative and quantitative evaluations are provided below; additional results are in the Appendix.

**Variant and hyperparameter analysis.** In Fig. 4(a), both random and contiguous strategies improve attention localization over dispersed patches, with contiguous selections preserving semantic coherence and random selections enhancing robustness through stochastic variability (see also Figs. 2 and 3 for comparisons of attention maps). Fig. 4(b) shows that low-pass filtering clearly outperforms high-pass, noise, or color jitter, as it suppresses irrelevant high-frequency details while retaining discriminative structure. Fig. 4(c)-(e) show that DEFOCA is robust to grid size, patch ratio, and blur strength, with performance stable across broad ranges.

Table 2: Comparison on UFGVR datasets. Results are in %. Avg: SoyAgeing subsets; Mean: all five datasets. DEFOCA-Tiny ViT achieves the highest accuracy except on SoyGlobal, where global, diffuse cues favor CLE-ViT slightly. Our model achieves this with just 21.20M parameters.

| Method | Cotton80 | SoyLocal | SoyGene | SoyGlobal | SoyAgeing | | | | | | Mean | #Params(M) |
|---|---|---|---|---|---|---|---|---|---|---|---|---|
| | | | | | R1 | R3 | R4 | R5 | R6 | Avg | | |
| ResNet34 (He et al., 2016) | 20.8 | 31.0 | 46.7 | 35.3 | 66.8 | 65.3 | 64.0 | 66.0 | 52.4 | 62.9 | 39.3 | 21.80 |
| ResNet50 (He et al., 2016) | 43.8 | 42.3 | 51.6 | 39.1 | 73.1 | 68.5 | 67.1 | 71.3 | 59.2 | 67.8 | 48.9 | 25.56 |
| Tiny ViT (Wu et al., 2022) | 67.5 | 52.0 | 64.8 | 54.1 | 82.4 | 81.9 | 80.2 | 82.0 | 69.5 | 79.2 | 63.5 | 21.20 |
| SIM-Trans-ViT(Sun et al., 2022) | 54.5 | 25.0 | 15.5 | 34.8 | 69.9 | 73.2 | 73.1 | 73.9 | 63.2 | 70.7 | 40.1 | 86.86 |
| CLE-ViT(Yu et al., 2023a) | 63.3 | 47.2 | 78.5 | 75.2 | 80.1 | 83.3 | 84.2 | 86.4 | 76.0 | 82.1 | 69.2 | 87.90 |
| Mix-ViT(Yu et al., 2023b) | 60.4 | 56.2 | 79.9 | 51.0 | 79.3 | 77.2 | 78.0 | 79.2 | 67.9 | 76.3 | 64.7 | 113.46 |
| ILA(Rios et al., 2024) | 55.4 | 50.8 | 62.2 | 58.1 | N/A | N/A | N/A | N/A | N/A | 75.0 | 60.3 | 86.86 |
| CSDNet-Swin-B(Fang et al., 2024b) | 67.9 | 60.5 | 86.8 | 76.1 | 83.8 | 85.1 | 85.1 | 84.8 | 76.8 | 83.1 | 74.8 | 87.90 |
| CLCA(Rios et al., 2025) | 67.8 | N/A | N/A | 58.2 | N/A | N/A | N/A | N/A | N/A | 88.3 | 71.4 | 86.86 |
| PEL-R50(Zhang et al., 2025) | 62.9 | 58.7 | 79.5 | 57.6 | 84.2 | 82.5 | 84.1 | 84.6 | 71.7 | 81.4 | 68.0 | 25.56 |
| PEL-D121(Zhang et al., 2025) | 63.3 | 59.3 | 81.5 | 60.6 | 85.9 | 83.5 | 86.3 | 84.4 | 71.4 | 82.3 | 69.4 | 7.98 |
| DEFOCA-R34 | 35.0 | 45.3 | 56.3 | 39.5 | 71.9 | 68.0 | 70.6 | 72.1 | 58.7 | 68.3 | 48.8 | 21.80 |
| DEFOCA-R50 | 47.9 | 52.2 | 71.4 | 50.4 | 77.0 | 74.5 | 74.0 | 78.1 | 64.2 | 73.6 | 59.1 | 25.56 |
| DEFOCA-Tiny ViT | 70.0 | 65.7 | 83.8 | 68.2 | 83.0 | 86.0 | 87.3 | 88.3 | 76.3 | 84.2 | 74.3 | 21.20 |

**Discussion on fine-grained benchmarks.** Compared to CNNs, DEFOCA consistently boosts accuracy, particularly when paired with Tiny ViT (Table 1). Notably, DEFOCA-Tiny ViT achieves competitive results on FGVC-Aircraft and CUB-200, and remains highly competitive on Stanford Cars and NABirds, closely matching strong attention-based models (*e.g.*, DCAL-R50+ViT and PEDTrans-ViT) while using only 21.2M parameters (21.2M *vs*. 112.42M and 86.86M, respectively). These results highlight the ability of our blur-to-focus mechanism to enhance discriminative localization while maintaining robustness across varied FGVR tasks. Fig. 5 shows that DEFOCA generates sharper, more localized attention on discriminative regions across five benchmarks, whereas baseline Tiny ViT often produces diffuse or background-focused attention.

**Discussion on ultra-fine-grained datasets.** Compared to CNN and ViT baselines, DEFOCA consistently improves accuracy. DEFOCA-Tiny ViT delivers strong overall performance, achieving competitive results on Cotton80, SoyLocal, SoyGene, and SoyAgeing (Table 2). Gains are especially notable on SoyGene and SoyAgeing, where subtle, localized discriminative cues highlight the blur-to-focus mechanism's ability to sharpen attention on fine details. SoyGlobal is an exception: CLE-ViT slightly outperforms DEFOCA, as its categories rely on broad, spatially diffuse cues where stochastic patch degradation is less effective. Despite this, DEFOCA-Tiny ViT achieves very competitive mean accuracy across all datasets, highlighting its robustness, adaptability, and effectiveness in capturing fine-grained cues. Fig. 6 shows that DEFOCA produces tighter, more separable class clusters with attention focused on discriminative regions, compared to the baseline.

## 4 CONCLUSION

We introduce DEFOCA, a lightweight and principled layer designed for FGVR. By applying localized low-pass filtering to stochastically selected regions of input, DEFOCA suppresses high-frequency noise while preserving subtle, label-critical cues, thereby enhancing both robustness and interpretability. Unlike conventional augmentation strategies or explicit attention mechanisms, DEFOCA is architecture-agnostic, supervision-free, and can be seamlessly integrated into existing networks as a drop-in module. Our theoretical analysis shows that DEFOCA maximizes the preservation of discriminative regions, minimizes representation drift, increases the signal-to-noise ratio of critical features, and provides a provable reduction of the generalization gap. Extensive experiments on fine-grained and ultra-fine-grained benchmarks validate these properties, consistently improving recognition accuracy and robustness. Future directions include adaptive variants of DEFOCA and potential extensions to multimodal or generative tasks.

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

# A APPENDIX

## A.1 RELATED WORK

**Fine-grained visual recognition.** FGVR requires distinguishing categories with high intra-class similarity and low inter-class variation(Ding et al., 2021), such as bird species, car models, or plant types. Traditional approaches that rely on global feature learning often fail to capture the subtle, localized cues defining these categories(Wu et al., 2019). More recent methods use part-based or attention-driven mechanisms to explicitly localize discriminative regions(Liu et al., 2021). While effective under controlled conditions, these approaches typically require strong supervision(Zhang et al., 2019) (*e.g.*, part annotations) or pre-trained detectors, and their performance degrades under pose variation, occlusion, or cluttered backgrounds. In contrast, the DEFOCA layer provides a model-agnostic, simple, and flexible mechanism that encourages networks to focus on informative, discriminative regions. By stochastically applying Gaussian blur to contiguous regions, DEFOCA balances local detail and global context, improves feature stability, and enhances robustness to occlusion and pose variation, all without additional supervision or architectural changes. This highlights its ability to generalize where conventional part-based or attention methods struggle.

**Data augmentation and regularization for FGVR.** Data augmentation is a common strategy to improve generalization in FGVR(Pu et al., 2023; Yuan et al., 2023). Standard techniques such as cropping, flipping, rotation, or color jitter are primarily designed for coarse object recognition; while they enhance robustness, they often disrupt the subtle, localized cues that are critical for fine-grained classification(Pu et al., 2023; Li et al., 2020). More recent strategies, including Mixup, CutMix, and attention-guided augmentation, attempt to address these challenges but either depend on specific architectures or fail to consistently preserve discriminative regions, limiting generalization(Huang et al., 2021; Imran & Athitsos, 2020). In contrast, the DEFOCA layer stochastically applies Gaussian blur to contiguous regions, selectively suppressing irrelevant or noisy content while retaining fine-grained features. This stochastic degradation naturally encourages the network to focus on informative regions, acting as a soft attention mechanism that diversifies training signals and improves robustness without sacrificing critical local detail. By preserving essential cues while regularizing the model, DEFOCA addresses limitations that prior augmentation methods cannot fully resolve.

**Attention and part-based mechanisms.** Attention mechanisms and part-based models aim to explicitly localize and weight discriminative object parts, with representative approaches including bilinear pooling, spatial transformers, and self-attention modules(Wu & Wang, 2017; Wu et al., 2019). While effective, these methods are often architecture-specific, computationally demanding, and heavily reliant on accurate part detection(Wu et al., 2019; Wu & Wang, 2017). They may also struggle to generalize under pose variation, occlusion, or cluttered backgrounds(Heo et al., 2022; Basly et al., 2023). In contrast, the DEFOCA layer implicitly guides attention by stochastically applying Gaussian blur to contiguous regions, encouraging the network to focus on informative, discriminative features without introducing additional architectural constraints. This stochastic degradation reduces dependence on explicit part detection or handcrafted attention modules, while

maintaining robustness and preserving fine-grained cues. By offering an implicit, flexible attention mechanism, DEFOCA provides a simpler and more generalizable alternative to conventional attention- or part-based approaches.

**Theoretical foundations in FGVR.** Most FGVR approaches focus primarily on empirical performance, providing limited theoretical justification(Dubey et al., 2018a; Demidov et al., 2023). Core aspects such as label safety, representation stability, and generalization are often overlooked, leaving models vulnerable to overfitting on spurious cues or background artifacts(Fang et al., 2024a; Zhang et al., 2024). Only a few works attempt to provide guarantees on how transformations or regularization strategies impact feature learning(Dubey et al., 2018a; Chapman et al., 2024). In contrast, the DEFOCA layer is theoretically grounded: it preserves label-critical regions (label safety), bounds representation drift, and maximizes the combinatorial probability of safe transformations through contiguous patch selection. By combining selective low-pass filtering with stochastic multi-view regularization, DEFOCA narrows the generalization gap in a principled manner, offering a reliable and interpretable alternative to existing FGVR methods. This theoretical perspective directly informs the design of DEFOCA and explains its robustness in challenging FGVR scenarios.

## A.2 PROOF OF LEMMA 1

*Proof.* By the Lipschitz property of $f_{\boldsymbol{\theta}}$, if the transformation $\mathcal{T}$ is label-safe (modifies only non-discriminative patches), the squared representation drift is bounded by

$$\|f_{\boldsymbol{\theta}}(\boldsymbol{x}) - f_{\boldsymbol{\theta}}(\boldsymbol{x}')\|_2^2 \leq (Ln)^2.$$

If at least one discriminative patch is altered, the drift can be substantially larger, up to $M^2$, reflecting disruption of critical information. Applying the law of total expectation over these two mutually exclusive events gives

$$\mathbb{E}_{\boldsymbol{x}' \sim \mathcal{T}(\boldsymbol{x})} \left[ \|f_{\boldsymbol{\theta}}(\boldsymbol{x}) - f_{\boldsymbol{\theta}}(\boldsymbol{x}')\|_2^2 \right] \leq P_{\text{safe}} (Ln)^2 + (1 - P_{\text{safe}}) M^2.$$

This formalizes the intuition that transformations preserving label-safety induce minimal representation drift, while violations produce potentially large deviations. $\square$

## A.3 PROOF OF PROPOSITION 1

*Proof.* By construction, the energy of discriminative patches (numerator) is unchanged, while the energy of non-discriminative patches (denominator) is attenuated by the Gaussian filter $H_{\sigma_i}(\omega)$.

Because $H_{\sigma_i}(\omega) < 1$ for high-frequency components, the denominator strictly decreases, and hence the SNR strictly increases.

This formalizes that DEFOCA enhances the relative contribution of discriminative information while suppressing irrelevant high-frequency noise. $\square$

## A.4 DATASETS AND STATISTICS

Experiments are conducted on widely used datasets. CUB-200-2011 contains 11,788 images of 200 bird species, following the standard train/test split. Stanford Cars includes 16,185 images of 196 car models, with 8,144 training and 8,041 test images. NABirds contains 48,562 images of 555 categories, with 23,938 training and 24,633 test images. FGVC-Aircraft comprises 10,000 images of 100 aircraft models, following the standard split. For ultra-fine-grained evaluation, we use the UFGVR dataset, which contains five major datasets: Cotton80, SoyLocal, SoyGlobal, SoyAgeing, and SoyGene. Statistics of the FGVR and UFGVR datasets used in our experiments are summarized in Tables 3 and 4.

## A.5 COMPLEXITY ANALYSIS

**Algorithm overview.** Given an input image of size $H \times W$ (total pixels $HW$), DEFOCA first partitions the image into a $P \times P$ grid, yielding $N = P^2$ patches. Let $s \in (0, N)$ denote the number of key patches and $n$ the number of selected patches, so that $n/N$ represents the processed pixel

Table 3: Overview of FGVR datasets used in our experiments.

| Dataset | Train | Test | Categories |
|---|---|---|---|
| CUB-200-2011 (Wah et al., 2011) | 5,994 | 5,794 | 200 |
| Stanford Cars (Krause et al., 2013) | 8,144 | 8,041 | 196 |
| NABirds (Van Horn et al., 2015) | 23,938 | 24,633 | 555 |
| FGVC Aircraft (Maji et al., 2013) | 3,333 | 3,333 | 100 |
| UFGVR collection (Yu et al., 2021b) | 24,367 | 22,747 | 3,526 |

Table 4: Overview of UFGVR datasets used in our experiments.

| Dataset | Train | Test | Categories | Train/Class |
|---|---|---|---|---|
| Cotton80 | 240 | 240 | 80 | 3.0 |
| SoyLocal | 600 | 600 | 200 | 3.0 |
| SoyGene | 12,763 | 11,143 | 1,110 | 11.5 |
| SoyAgeing | 4,950 | 4,950 | 198 | 25.0 |
| SoyGlobal | 5,814 | 5,814 | 1,938 | 3.0 |

fraction. Each patch has width $p_w = \lfloor W/P \rfloor$ and height $p_h = \lfloor H/P \rfloor$, with border patches in the last row and column absorbing residual pixels to ensure exact tiling.

Patch indices are then chosen according to a placement strategy. In the *random* strategy, indices are sampled without replacement from the $N$ candidates. In the *contiguous* strategy, selection begins from a random seed and expands to neighboring patches until $n$ are obtained, with a fallback to random sampling if expansion stalls. The *dispersed* strategy instead applies a farthest-point heuristic, adding at each step the patch that maximizes the minimum squared grid distance to the set of already chosen patches.

For each selected index $i \in S$, the corresponding patch is cropped, a per-patch operator is applied, and the result is pasted back. The default operator is Gaussian blur with fixed standard deviation $\sigma$, though any user-supplied operator `patch_op` can be substituted. In the multi-view variant, Multiview-DEFOCA generates $V > 1$ distinct views by repeatedly sampling unique index sets (up to a user-defined maximum number of attempts) and applying the same crop-process-paste pipeline. If a post-processing transform is provided, the resulting $V$ views are stacked into a tensor of shape $(V, C, H, W)$.

Unless otherwise stated, several assumptions hold throughout. Gaussian blur is implemented with fixed kernel cost proportional to the number of pixels, any custom per-patch operator has bounded per-pixel complexity, and standard image manipulations such as crop, or paste are linear in the number of affected pixels. Finally, uniqueness checks rely on hash sets with amortized $O(1)$ cost.

### A.5.1 TIME AND SPACE COMPLEXITY

**Notation.** We denote $N = P^2$ as the total number of patches in a $P \times P$ grid, with $n \in (0, N)$ the number of selected patches. The processed fraction is $n/N$. The input image has width $W$, height $H$, channels $C$, and $V$ views are generated in the multi-view variant. Asymptotic behavior is expressed in big-$O$ and big-$\Theta$ notation.

**Selection cost.** The cost of index selection depends on the placement strategy. Random sampling requires $O(n)$ operations by drawing $n$ indices without replacement from $N$. Contiguous placement is also $O(n)$, since each newly added patch comes from a constant number of neighbors (at most four) of the growing region. The dispersed strategy is more expensive: using a greedy farthest-point heuristic, each of the $n - 1$ iterations scans the remaining $N - k$ candidates against $k$ selected patches, yielding $\sum_{k=1}^{n-1}(N - k)k = \Theta(nN)$ total comparisons.

**Per-patch application cost.** Once indices are chosen, $n$ patches are processed. Each patch covers $(WH)/N$ pixels, so the total number of affected pixels is $(n/N)WH$. With operators such as Gaussian blur (fixed $\sigma$) or any bounded per-pixel operator, this leads to $\Theta((n/N)WH)$ cost. Additional overhead from cropping, pasting, or drawing borders is only a constant factor.

Table 5: Time complexity of different components.

| Component | Time Complexity |
|---|---|
| DEFOCA (random/contiguous) | $\Theta(\frac{n}{N}WH) + O(n)$ |
| DEFOCA (dispersed) | $\Theta(\frac{n}{N}WH) + O(nN)$ |
| Multiview DEFOCA (random/contiguous) | $O(\texttt{max-attempts} \cdot n) + \Theta(V\frac{n}{N}WH)$ |
| Multiview DEFOCA (dispersed) | $O(\texttt{max-attempts} \cdot nN) + \Theta(V\frac{n}{N}WH)$ |

Table 6: Peak space complexity of different components.

| Component | Space Complexity (Peak) |
|---|---|
| DEFOCA (random/contiguous) | $\Theta(WH) + O(n)$ |
| DEFOCA (dispersed) | $\Theta(WH) + O(P)$ |
| Multiview DEFOCA (random/contiguous) | PIL: $\Theta(VWH)$; Tensor: $\Theta(VCWH) + O(Vn)$ |
| Multiview DEFOCA (dispersed) | Same as above |

**Single-view complexity.** For one application of DEFOCA, the runtime is

$$\Theta\left(\tfrac{n}{N}WH\right) + \text{SelectCost},$$

where SelectCost $= O(n)$ for random or contiguous placement and SelectCost $= O(nN)$ for dispersed placement. Peak memory is dominated by the image copy $\Theta(WH)$ with negligible overhead from index bookkeeping.

**Multi-view and uniqueness.** In Multiview-DEFOCA, up to $\texttt{max-attempts}$ index selections may be performed in order to collect $V$ unique sets, each attempt costing SelectCost. For each accepted set, the patch-application stage costs $\Theta((n/N)WH)$. The overall runtime is therefore

$$O(\texttt{max-attempts} \cdot \text{SelectCost}) + \Theta\left(V\,\tfrac{n}{N}WH\right).$$

In practice, when collisions are rare, the second term dominates since $\texttt{max-attempts}$ is $O(V)$.

For memory, the peak depends on the output format. If views are kept as PIL images, the space is $\Theta(VWH)$. If they are converted to tensors, the stacked output has size $\Theta(VCWH)$, with an additional $O(Vn)$ overhead for index sets. Temporary duplication during stacking only changes constants.

**Implementation notes.** Border patches (last row or column) absorb extra pixels but do not affect asymptotic orders. If Gaussian kernel cost scales with $\sigma$, this introduces only a constant multiplicative factor. In regimes with very large $P$ and $V$ but small $n$, uniqueness constraints may cause $\texttt{max-attempts}$ to bind; the bound above already accounts for this worst case.

A.5.2  EMPIRICAL RUNTIME TRENDS AND DISCUSSION

The empirical runtime plots in Figs. 7 and 8 align closely with our theoretical complexity analysis in §A.5.1. As anticipated, the *dispersed* strategy incurs the greatest computational overhead, consistent with its $O(nN)$ selection complexity. Empirically, this appears as a visibly super-linear increase in runtime as both the grid resolution $N$ and the active fraction $n/N$ grow. The *dispersed* curves are not only consistently higher than those for *random* and *contiguous*, but also exhibit an accelerated growth pattern that becomes increasingly pronounced at larger $n/N$. This trend can be intuitively explained by the farthest-point heuristic: each new patch selection requires scanning a large candidate set, and as the density of processed patches rises, this repeated scanning amplifies the cost.

In contrast, the *random* and *contiguous* strategies, both bounded by $O(n)$ selection, remain effectively linear in $n$. Their runtime is dominated by per-pixel operations, leading to nearly linear growth in practice. Even at the largest tested grid sizes, their curves maintain a gentle slope that agrees with theory. Between these two, *contiguous* placement shows slightly higher overhead than *random*, due to the bookkeeping required for neighbor expansion, but the gap remains small compared to the steep divergence of *dispersed*.

Taken together, these results demonstrate a strong correspondence between asymptotic predictions and empirical scaling. When $n/N$ is small, all strategies perform comparably. As $n/N$ increases,

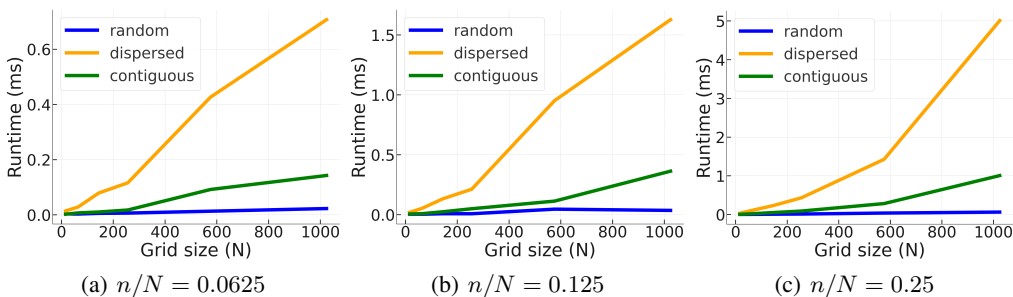

(a) $n/N = 0.0625$  (b) $n/N = 0.125$  (c) $n/N = 0.25$

Figure 7: Runtime (ms) versus selection ratio $n/N$ across placement strategies. The selection ratio $n/N$ denotes the fraction of patches processed. The three plots illustrate how runtime scales under different placement strategies (contiguous, dispersed, random) as the selection ratio increases.

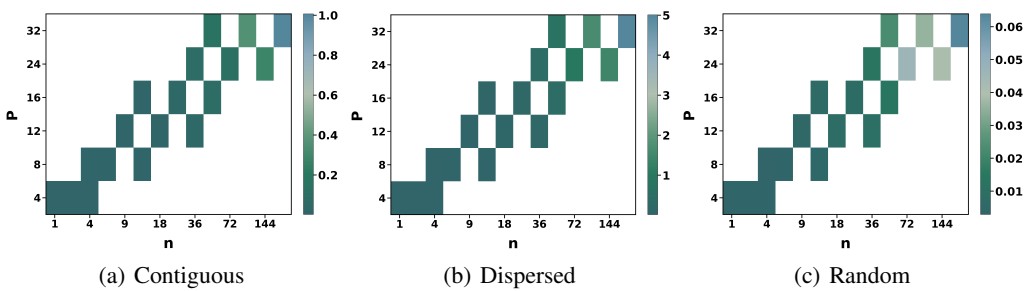

(a) Contiguous  (b) Dispersed  (c) Random

Figure 8: Runtime (ms) across placement strategies, number of selected patches $n$, and grid divisions $P$. The figure compares the runtime of different patch placement strategies (contiguous, dispersed, random) as both the number of selected patches $n$ and the grid resolution $P$ increase.

however, the theoretical gap becomes visible: *dispersed* grows much faster, roughly quadratic in $n/N$ relative to $n$, while *random* and *contiguous* remain close to linear. This confirms that although *dispersed* provides superior spatial coverage, it does so at a significant runtime cost, which can become prohibitive in high-resolution or high-coverage regimes.

### A.5.3  TRAINING AND INFERENCE COST

**Training cost.** We separate the cost into two components: (i) runtime (wall-clock time) and (ii) GPU memory (VRAM).

*Runtime.* The cost arises mainly from two factors. The first is patch selection and processing: an image is divided into $N = P \times P$ patches, of which a fraction $n/N$ is modified. The dominant work here is $\Theta(n/N \times HW)$ pixel operations, plus a small index-selection overhead (linear for random or contiguous placement). This term is usually negligible unless both $P$ and $n/N$ are large. The second, and more significant, factor is *multi-view augmentation*. Generating $V \geq 1$ augmented views per image increases the number of forward and backward passes approximately linearly with $V$, leading to near-linear per-epoch scaling: $T_{\text{epoch}}(V) \approx V \cdot T_{\text{epoch}}(1)$, up to minor constant factors.

*GPU memory.* The effective activation footprint scales with the product of batch size $B$ and the number of views $V$: VRAM $\propto B \times V$, in addition to the fixed model and optimizer overhead. Thus, under a fixed memory budget, batch size and number of views can be traded off, *e.g.*, $(B = 32, V = 1) \approx (B = 16, V = 2) \approx (B = 8, V = 4)$.

In practice, mixed precision training and gradient accumulation further mitigate these trade-offs, allowing efficient scaling. Additional empirical relationships are illustrated in Fig. 9.

**Inference cost.** At test time, the DEFOCA layer is disabled, and inference proceeds with a standard single-view forward pass. As a result, FLOPs, latency, and memory usage remain identical to the backbone model. This zero-overhead inference aligns with sustainable computing goals (*e.g.*,

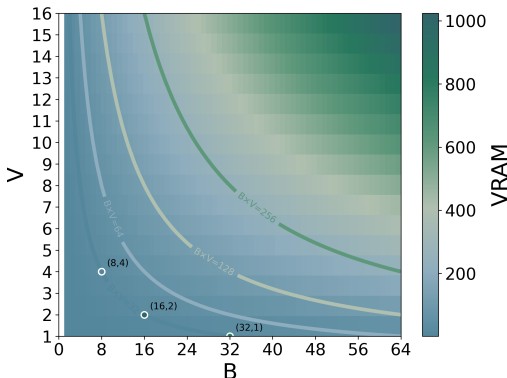

Figure 9: VRAM trade-off between batch size $B$ and number of views $V$. The heatmap shows relative activation memory $\propto B \times V$; iso-VRAM contours (*e.g.*, $B \times V = 64, 128, 256$) indicate constant memory budgets, with example configurations marked.

Table 7: Data efficiency evaluation with 5 training images per class. Accuracy gains are measured relative to the baseline backbone. Efficiency $\eta_{\text{eff}}$ quantifies improvement per million parameters. Backbone parameter counts: EfficientNet (B0 version) 5.29M; ResNet50 25.56M; ResNet34 21.80M. pp = percentage points.

| Method | Food101@5 | OxfordIIITPet@5 | Flowers102@5 | $\eta_{\text{eff}}$(pp/M) |
|---|---|---|---|---|
| ResNet34 | 23.2% | 73.1% | 77.0% | – |
| DEFOCA-R34 (Ours) | **24.3%** | **73.8%** | **80.9%** | **0.09** |
| ResNet50 | 27.1% | 78.6% | 82.4% | – |
| DEFOCA-R50 (Ours) | **34.2%** | **79.1%** | **83.1%** | **0.11** |
| EfficientNet_B0 | 26.8% | 64.4% | 84.0% | – |
| DEFOCA-EfficientNet_B0 (Ours) | **30.4%** | **64.8%** | **85.4%** | **0.34** |

SDG 7: affordable and clean energy), as any additional training-time cost does not increase energy demand during deployment.

### A.5.4 REAL-WORLD DEPLOYMENT AND EDGE COMPUTING

The proposed DEFOCA layer integrates seamlessly into standard training pipelines and is particularly well-suited for resource-constrained scenarios, such as smart agriculture or edge devices. It introduces no architectural changes and adds no parameters. Importantly, at test time, the layer is disabled, so FLOPs, latency, and memory usage are identical to the backbone model. We plan to release the code upon acceptance.

To demonstrate edge-friendliness and data efficiency, we conducted a study restricting the training set to only five images per class while keeping the test sets unchanged. Table 7 summarizes the results on three datasets: Food101 (Bossard et al., 2014), Oxford-IIIT Pet (Parkhi et al., 2012), and Flowers102 (Nilsback & Zisserman, 2008). We also report an efficiency indicator, $\eta_{\text{eff}} = \Delta_{\text{avg}}/\text{Params(M)}$, defined as the mean accuracy gain (percentage points) across the three datasets divided by the backbone parameter count in millions. This metric highlights performance improvements relative to model size, which is critical for edge deployment.

Normalizing gains by parameter count highlights DEFOCA's suitability for edge devices. Compact backbones like EfficientNet (B0 version) achieve the highest efficiency (0.34 pp/M), while larger models such as ResNet50 and ResNet34 also see consistent improvements (0.11 and 0.09 pp/M, respectively). This shows that DEFOCA reliably enhances accuracy without increasing computational cost at test time, making it particularly attractive for deployment in resource-limited settings.

### A.6 EXPERIMENTAL SETUP

Experiments were conducted on NVIDIA GPUs. Smaller models, such as ResNet34 and ResNet50, were trained on V100 (32GB, SXM) nodes, while larger DEFOCA configurations used A100 (80 GB) nodes. V100 and A100 nodes provided 12 and 16 CPU cores, respectively, with system memory set to 24 GB for most datasets and 60 GB for NABirds, SoyGene, and SoyGlobal. Datasets were loaded into host memory at the start of each run, and all training was performed on a single GPU without distributed execution.

The software stack included Rocky Linux 8, Python 3.11.13 (uv 0.7.19), GCC 8.5.0, and CUDA 12.6.2. Key packages were pinned for reproducibility: torch 2.8.0, timm 1.0.19, opencv-python 4.11.0.86, pillow 11.3.0, and numpy 2.3.2. All training used FP32 precision without AMP, BF16, or TF32.

Full environment specifications and scripts will be released with the code.

### A.7 LIMITATIONS AND FUTURE WORKS

**Limitations.** While DEFOCA consistently improves fine- and ultra-fine-grained recognition, we observed modest gains on datasets where class cues are comparatively global or spatially diffuse, such as Stanford Cars and FGVC-Aircraft. In these cases, randomly or contiguously blurring local patches removes little nuisance signal and can occasionally attenuate class evidence spread across large structures, such as body contours or wing profiles, reducing the margin over strong baselines. This aligns with our ultra-fine-grained study, where global regularities, such as domain-level texture or style, dominate, making stochastic local defocus less advantageous.[2]

Our current instantiation depends on implicit assumptions about key regions. The patch grid size $P \times P$, number of processed patches $n$, and single blur scale $\sigma$ per view assume a roughly constant number of key regions per image and comparable sizes across images. When discriminative evidence varies widely in count or extent, such as a single large grille versus many small emblems, a fixed $(P, n, \sigma)$ can either under-cover, leaving spurious background intact, or over-degrade, affecting informative areas.

DEFOCA is also sensitive to capture scale and framing. It is most label-safe when the object of interest occupies a moderate field of view and key parts are compact. Extreme zoom-in/zoom-out or tight crops reduce the separation between discriminative and non-discriminative patches, increasing the likelihood that blur overlaps informative pixels. This explains smaller gains on datasets where the canonical viewpoint emphasizes large, globally informative shapes.

Placement strategies involve a trade-off with computational cost. Dispersed placement improves coverage of diffuse cues but is more expensive to sample than random or contiguous strategies. In practice, this limits the number of views $V$ during training, which in turn caps the regularization benefit DEFOCA can provide on high-resolution inputs.

Finally, our theoretical analysis relies on idealized assumptions. It assumes the existence of a small set $S$ of discriminative patches, that $s \ll N$, and Lipschitz stability of $f_{\boldsymbol{\theta}}$ to non-critical patch perturbations. In problems where cues are broad-band or $S$ is large or unknown, the combinatorial label-safety advantage of structured placement diminishes and the drift bounds become less tight.

**Future works.** Future directions for DEFOCA focus on making the method more adaptive, task-aware, and efficient. One avenue is to replace hand-crafted placement policies with a data-driven controller that predicts per-image blur masks from cheap signals, such as early-layer saliency, gradient variance, or uncertainty. Such a controller could avoid high-saliency regions to preserve label-critical details while targeting backgrounds and repetitive textures. A curriculum that starts with dispersed placement and gradually shifts toward contiguous patterns could further stabilize training.

Extending patching and blur to multiple scales offers another opportunity. Heterogeneous patch sizes and per-patch blur strength can handle variability in key-region size and capture-scale shifts. Mixed pyramids with coarse patches for global clutter and fine patches near likely discriminative parts, combined with differentiable, learnable kernels such as separable Gaussians or anisotropic

---

[2]See our ablations on patch selection strategies and operations, along with the main evaluation tables. DE-FOCA is most effective when discriminative cues are spatially sparse and localized.

filters, could adaptively adjust blur based on local frequency content. Beyond simple low-pass filters, task-aware operators such as band-pass suppression of repetitive backgrounds, structure-preserving smoothing, or frequency-domain mixing may improve performance on global-cue datasets.

Hybrid placement strategies can further enhance flexibility. For categories with diffuse cues, such as cars or aircraft, combining dispersed coverage with one or two contiguous blobs can preserve semantic coherence while capturing distributed evidence. Data-driven estimation of label safety using proxies such as teacher-student consistency or holdout agreement could penalize transformations that alter pseudo-labels, creating a closed-loop safe defocus module that maximizes coverage without harming critical features.

At test time, defocus could be used to improve robustness. Ensembling a few defocused views or applying targeted test-time blur when model confidence is low may help handle occlusions and clutter. Extensions to video and multi-modal tasks also hold promise. Temporal coherence in video can regularize over occlusion and pose jitter, while in vision-language settings, blur masks could align with token-level attributions to guide which regions are preserved.

Practical implementation and broader evaluation are key. Efficient placement can be optimized for hardware using separable convolutions, tiled kernels, cached integral images, or low-rank updates. Hybrid supervision from weak part cues can improve label safety without complicating the architecture. Stress-testing adaptive variants on scale and framing shifts, as well as global-cue benchmarks, with diagnostic metrics for coverage, drift, and signal-to-noise ratio changes, will provide a fuller picture of robustness and effectiveness.

## A.8 ADDITIONAL EVALUATIONS

We compare DEFOCA with additional contrastive learning and well-known methods on the UFGVR datasets in Table 8.

**Discussions.** Overall, DEFOCA-Tiny ViT consistently outperforms both CNN- and ViT-based baselines, demonstrating its ability to capture subtle, localized discriminative cues that are critical in these datasets. On Cotton80 and SoyLocal, DEFOCA achieves gains of over 6-9 percentage points compared to the strongest prior methods, highlighting its effectiveness when discriminative features are spatially sparse and fine-grained.

The performance advantage is particularly pronounced on SoyGene and SoyAging, where subtle differences in leaf texture or growth patterns are key for classification. Traditional attention-based methods such as ADL and Hide-and-Seek, as well as standard contrastive learning techniques like SimCLR and MoCo, show limited improvements on these datasets, likely because their global or uniform augmentation strategies fail to emphasize such localized features. In contrast, DEFOCA's stochastic blur-to-focus mechanism guides the model to concentrate on informative regions while regularizing the rest, resulting in tighter, more discriminative feature representations.

On datasets like SoyGlobal, where class cues are more diffuse and broadly distributed, the gains are less extreme but still significant. While CLE-ViT achieves the highest accuracy here, DEFOCA remains highly competitive, closely matching its performance despite having fewer parameters and a simpler architecture. This illustrates the robustness and adaptability of DEFOCA: it excels in scenarios with both highly localized cues and moderately distributed features, without requiring task-specific tuning or architectural modifications.

Comparing ViT backbones with CNN-based approaches, it is evident that transformer architectures benefit substantially from DEFOCA, as the patch-based stochastic attention aligns well with the intrinsic tokenization in ViTs. Even the Tiny ViT variant, with a compact parameter footprint, consistently surpasses larger CNN and hybrid models, reinforcing the method's efficiency and suitability for resource-constrained or real-world deployment. Overall, these results confirm that DEFOCA effectively enhances feature focus, improves class separability, and delivers robust performance across diverse ultra-fine-grained recognition tasks.

Table 8: Top-1 accuracy (%) comparison across methods.

| Method | Backbone | Cotton80 | SoyLocal | SoyGene | SoyAging | SoyGlobal |
|---|---|---|---|---|---|---|
| SimCLR (FT) (Chen et al., 2020a) | ResNet50 | 51.67 | 37.33 | 62.68 | 64.73 | 42.54 |
| SimCLR (L) (Chen et al., 2020a) | ResNet50 | 41.25 | 29.17 | 29.62 | 46.18 | 13.48 |
| MoCo v2 (FT) (Chen et al., 2020b) | ResNet50 | 45.00 | 32.67 | 56.49 | 59.13 | 29.26 |
| MoCo v2 (L) (Chen et al., 2020b) | ResNet50 | 30.42 | 27.67 | 26.58 | 38.26 | 12.99 |
| BYOL (FT) (Grill et al., 2020) | ResNet50 | 52.92 | 33.17 | 60.65 | 64.75 | 41.35 |
| BYOL (L) (Grill et al., 2020) | ResNet50 | 47.92 | 25.50 | 35.13 | 49.53 | 18.44 |
| Cutout (8) (DeVries & Taylor, 2017) | ResNet50 | 55.83 | 37.67 | 61.12 | 65.70 | 47.06 |
| Cutout (16) (DeVries & Taylor, 2017) | ResNet50 | 54.58 | 31.67 | 62.46 | 63.68 | 44.65 |
| Hide and Seek (Kumar Singh & Jae Lee, 2017) | ResNet50 | 48.33 | 28.00 | 61.27 | 60.48 | 23.74 |
| ADL (0.5) (Choe & Shim, 2019) | ResNet50 | 43.75 | 34.67 | 55.19 | 61.70 | 39.35 |
| ADL (0.25) (Choe & Shim, 2019) | ResNet50 | 40.83 | 28.00 | 52.18 | 51.56 | 29.50 |
| Cutmix (Yun et al., 2019) | ResNet50 | 45.00 | 26.33 | 66.39 | 62.68 | 30.31 |
| DCL (Chen et al., 2019) | ResNet50 | 53.75 | 45.33 | 71.41 | 73.19 | 42.21 |
| MaskCOV (Yu et al., 2021a) | ResNet50 | 58.75 | 46.17 | 73.57 | 75.86 | 50.28 |
| SPARE (Yu et al., 2022) | ResNet50 | 60.42 | 44.67 | 79.41 | 75.72 | 56.45 |
| ViT (Dosovitskiy et al., 2020) | ViT | 52.50 | 38.83 | 53.63 | 66.95 | 40.57 |
| DeiT (Touvron et al., 2021) | ViT | 54.17 | 38.67 | 66.80 | 69.54 | 45.34 |
| TransFG (He et al., 2022a) | ViT | 54.58 | 40.67 | 22.38 | 72.16 | 21.24 |
| Hybrid ViT (Dosovitskiy et al., 2020) | ViT+R50 | 50.83 | 37.00 | 71.74 | 73.56 | 18.82 |
| Mix-ViT (Yu et al., 2023b) | ViT+R50 | 60.42 | 56.17 | 79.94 | 76.30 | 51.00 |
| CLE-ViT (Yu et al., 2023a) | ViT | 63.33 | 47.17 | 78.50 | 82.14 | **75.21** |
| DEFOCA-Tiny ViT | Tiny ViT | **70.0** | **65.7** | **83.8** | **84.18** | 68.2 |

## A.9 ADDITIONAL VISUALIZATIONS

### A.9.1 $P_{\text{SAFE}}$ VISUALIZATIONS

To make the label-safety formula more intuitive, we provide two illustrative figures.

In Fig. 10, we observe that if $n/N$ does not exceed 0.2, the label-safety probability remains at least 0.6, and when $n/N$ is below 0.1, it exceeds 0.8.

Fig. 11 shows that for a fixed $P = 5$ and $s \in \{4, 8, 12, 16, 20, 24\}$, the label-safety probability drops sharply as the ratio increases.

In practice, we recommend starting with a small $P \in \{4, 5, 6\}$ if improvements are minor, and then exploring $P \in \{10, 15, 20\}$ to potentially achieve higher theoretical accuracy.

### A.9.2 T-SNE VISUALIZATIONS FOR CUB, AIRCRAFT AND STANFORD CARS

Additional t-SNE maps are provided for further comparison.

In Fig. 12, it can be observed that without DEFOCA, the representation space appears more entangled, with certain classes overlapping and forming indistinct clusters. In contrast, with DEFOCA, the class boundaries become more clearly separated.

In Fig. 13, although the learned space without DEFOCA demonstrates nearly perfect cluster boundaries, some samples from the "Parakeet Auklet" and "Rhinoceros Auklet" species remain mixed together. With DEFOCA, this issue is largely reduced, and the learned space shows more distinct and well-formed clusters.

In Fig. 14, without DEFOCA, the representation space is highly tangled, whereas with DEFOCA, the class separation becomes noticeably clearer.

Summarily, these three t-SNE visualizations reveal how DEFOCA influences the learned space, enabling us to directly observe which types of features are improved and which classes benefit most from the enhancement.

### A.9.3 SILHOUETTE VISUALIZATIONS FOR CUB, AIRCRAFT AND STANFORD CARS

We provide the Silhouette score to evaluate each t-SNE space, and visualize the results in a chart for an intuitive comparison, allowing a clearer understanding of the detailed gaps between the t-SNE maps between the DEFOCA and the baseline t-SNE maps.

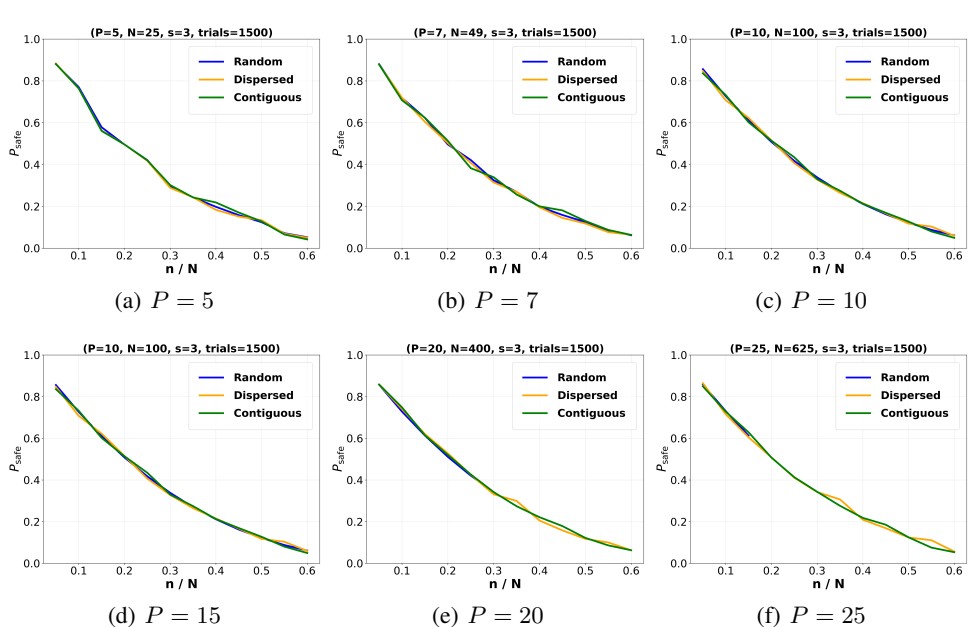

Figure 10: Label-safety probability (Assume $s = 3$).

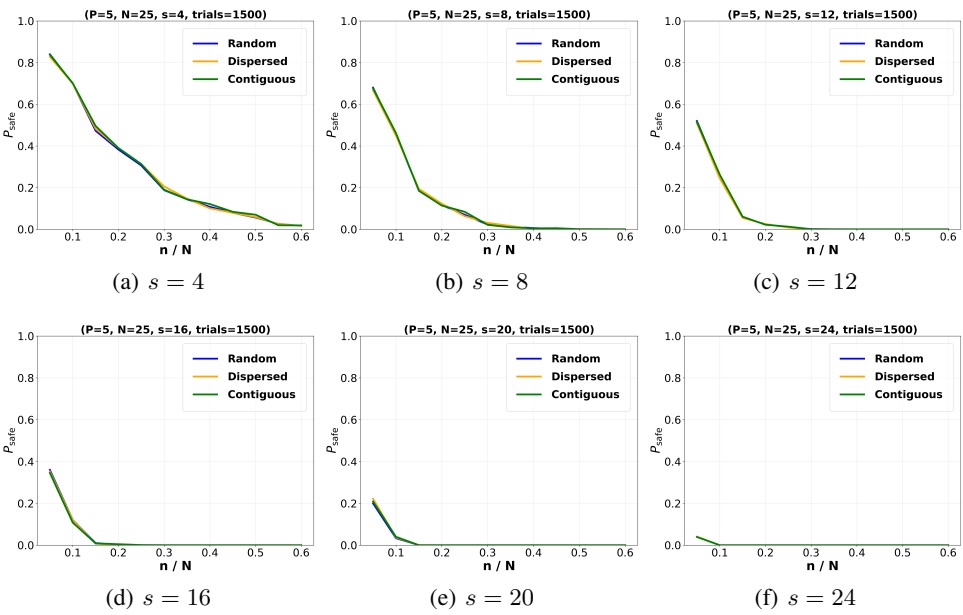

Figure 11: Label-safety probability ($P = 5$).

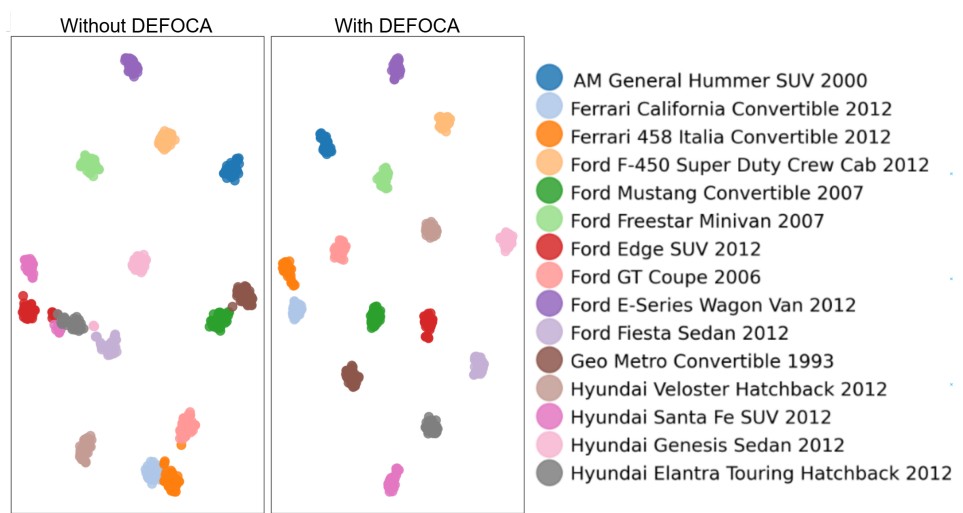

Figure 12: t-SNE visualization comparison: Stanford Cars.

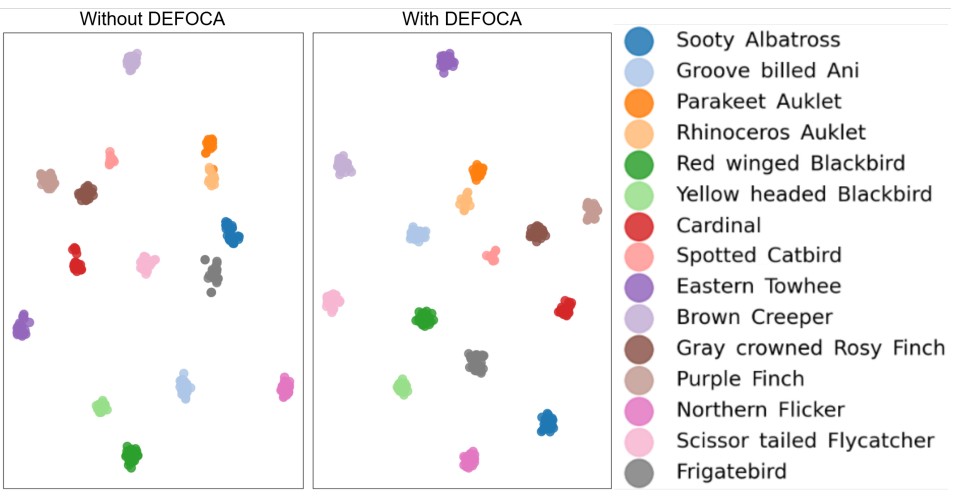

Figure 13: t-SNE visualization comparison: CUB-200-2011.

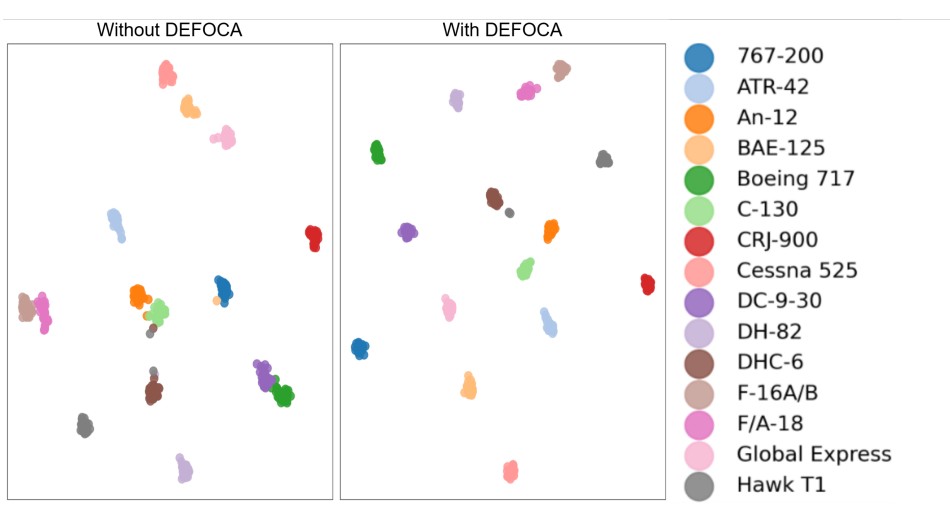

Figure 14: t-SNE visualization comparison: FGVC-Aircraft.

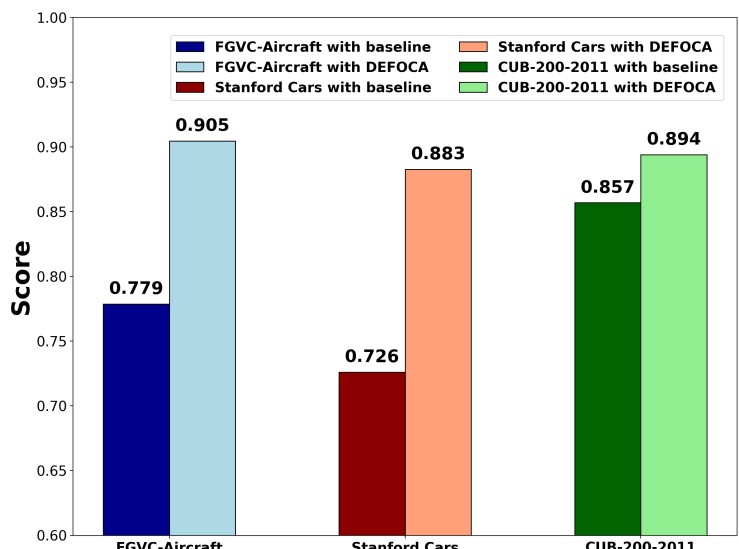

Figure 15: Silhouette visualization comparison: Stanford Cars, CUB-200-2011, FGVC-Aircraft.

In Fig. 15, a significant trend is demonstrated: all methods with DEFOCA achieve higher scores than those without DEFOCA. The largest gap is observed in Stanford Cars, with a difference of 0.157, while the most negligible improvement appears in CUB-200-2011, only improved 0.037.

Overall, DEFOCA shows meaningful enhancement across datasets and demonstrates consistency in the discriminative capability of the learned space.

### A.10    LLM USAGE DECLARATION

We disclose the use of Large Language Models (LLMs) as general-purpose assistive tools during the preparation of this manuscript. LLMs were used only for minor tasks such as grammar and style improvement, code verification, and formatting suggestions. No scientific ideas, analyses, experimental designs, or conclusions were generated by LLMs. All core research, methodology, experiments, and results were performed and fully verified by the authors.

The authors take full responsibility for all content presented in this paper, including text or code suggestions that were refined with the assistance of LLMs. No content generated by LLMs was treated as original scientific work, and all references and claims have been independently verified. LLMs did not contribute in a manner that would qualify them for authorship.

