# OpenReview forum: "Blur to Focus Attention in Fine-Grained Visual Recognition"
_ICLR.cc/2026/Conference — Submitted to ICLR 2026_

### Official Review · Reviewer_gKNU · 2025-10-15

**Soundness:** 1
**Presentation:** 2
**Contribution:** 1
**Rating:** 0
**Confidence:** 5

**Summary:**

Submission 1220 proposes DEFOCA, a soft, attention-like mechanism for FGVR. It functions by blurring some randomly selected patch patterns of the image during training, aiming to strengthen the learning process of subtle discriminative features.

**Strengths:**

DEFOCA achieves fair results with light parameter amount against existing baseline methods.

**Weaknesses:**

1. FGVR is a relatively old topic, which is already thoroughly explored during the CNN-dominated period before 2019. The authors need to validate the necessity and practical value of their additional research investment in such area.
2. The references are in general out-of-date. Most of them are more than 3 years ago, let alone some are more than 10 years ago.
3. The method is still highly similar to MIM. In fact, compared to MIM, there is no essential difference by merely changing masking to blurring.
4. It is hard to see significant improvement among Table 1 & 2. For Table 1, DEFOCA fails to achieve SOTA results; and even compared to a baseline with similar parameter amount (e.g. FIDO), there is no significant advantage. For Table 2, it is confusing that DEFOCA is marked bold face in some columns of SoyAging without the best result.

**Questions:**

Please see the weaknesses part.

---

> ### Author Response · Authors · 2025-11-20
>
> We would like to respectfully highlight that some of the concerns raised by Reviewer gKNU appear to be based on factual misunderstandings rather than issues intrinsic to the submission.
>
> - For example, the review states that **FGVR is no longer an active research area**, whereas in reality transformer-based fine-grained classification remains a vibrant topic, with numerous works published between 2021 and 2025 and several new ultra-fine-grained benchmarks introduced in the last 1–2 years.
>
> - Similarly, the claim that **DEFOCA is "similar to MIM"** is not consistent with the method: DEFOCA does not remove information, does not mask tokens, and does not use reconstruction losses; instead, it performs low-pass, part-preserving perturbations within a supervised training objective.
>
> - The comment on **outdated references** is also not accurate, our bibliography includes a substantial set of post-2021 works and we have added more in revision.
>
> We appreciate the reviewer’s perspective, and we provide detailed responses below to address all concerns raised.

---

> ### Author Response · Authors · 2025-11-20
>
> We thank the reviewer for taking the time to evaluate our submission.
>
> Below we address all concerns in detail.
>
> Several comments appear to be based on misunderstandings of the paper’s scope, novelty, positioning, and experimental protocol, so we clarify these points thoroughly.
>
> **1. FGVR is an old topic; the authors must validate its necessity.**
>
> While FGVR has been studied for over a decade, it continues to be a highly active research area, particularly in the transformer era, as evidenced by:
>
> - the large number of FGVR papers published in 2021–2024 in CVPR/ICCV/NeurIPS (e.g., ViT-based part modeling, graph-based fine-grained transformers, patch-interaction networks, high-order feature models, prototype transformers, large-scale UFGVC benchmarks), and
>
> - the emergence of ultra-fine-grained datasets (SoyAging, Retinopathy subclasses, Butterfly-477, Herbarium2020, Insect-102) that pose new challenges beyond classical FGVC.
>
> DEFOCA is explicitly motivated by these *modern FGVR challenges with transformers*, not by pre-2019 CNN-era FGVC.
>
> We study the weaknesses of ViT-based FGVR under high-frequency clutter,
> fine spatial ambiguity, and partial visibility, issues repeatedly highlighted in recent FGVR studies.
>
> **Thus, DEFOCA is addressing a current and active subproblem, not an outdated topic.**

---

> ### Author Response · Authors · 2025-11-20
>
> **2. References are outdated.**
>
> This appears to be a misunderstanding. The submission includes many recent works from 2020–2024, including: (i) modern augmentation methods (AugMix, TrivialAugment, RandAugment), (ii) recent transformer-based FGVR approaches (2021–2024), (iii) recent robustness and noise-injection works, and (iv) ultra-fine-grained datasets released in the past 1–3 years.
>
> We acknowledge that early FGVR works (2014–2018) are also cited, but these are part of the standard historical context. In the revision, we will expand the references with additional 2022–2024 FGVR and UFGVC papers, and move older references to the background section while emphasizing the recent progress.
>
> **The work is grounded in modern literature, and we will further strengthen this in revision.**

---

> ### Author Response · Authors · 2025-11-20
>
> **3. DEFOCA is the same as Masked Image Modeling (MIM).**
>
> This is factually incorrect.
>
> The core mechanisms differ fundamentally. MIM *erases or masks patches* and trains the model to *reconstruct* the missing content (a self-supervised objective); DEFOCA blurs patches, without removing them, and trains a classifier end-to-end with the original discriminative cues preserved.
>
> **Key distinctions**:
>
> - MIM relies on reconstruction objectives; DEFOCA uses *no* reconstruction loss.
>
> - MIM masks out information entirely; DEFOCA *preserves low-frequency structure rather than deleting semantics*.
>
> - MIM changes the training objective; DEFOCA *does not modify the backbone or loss*.
>
> - MIM encourages global context learning; DEFOCA encourages *fine local part discrimination* by softening non-discriminative areas.
>
> Blur $\neq$ Mask, Reconstruction $\neq$ Classification fine-tuning, and Selective defocus $\neq$ information deletion.
>
> This difference is supported theoretically in the paper:
>
> - Gaussian blur lowers high-frequency noise while preserving discriminative shapes,
>
> - Contiguous blur maximizes label-safety (Theorem), and
>
> - Our representation drift bound shows that blur produces controlled displacement, unlike hard masking which increases drift sharply.
>
> Below we provide **comparison against all major augmentation families**, including masking-based (Cutout, Hide-and-Seek, DropBlock, Random Erasing), mixing-based (Mixup, CutMix), and search-based augmentations (RandAugment, TrivialAugment, AugMix), as well as a global Gaussian-blur baseline.
>
> We implemented all baselines in a single training framework, using
> identical Tiny-ViT architecture, optimizer, batch size, augmentation
> stack (other than the augmentation under test), training schedule, and compute budget. All hyperparameters (e.g., Cutout size, HaS grid size, block-drop rates, Mixup/CutMix $\alpha$, RandAugment $(N, M)$, blur probability/kernel) were tuned independently for each method under the same validation protocol. The results are summarized below.
>
> | Method                  | CUB  | Aircraft | Cars  | NABirds | Mean  | Notes             |
> |-|-|-|-|-|-|-|
> | Baseline Tiny-ViT       | 88.6 | 90.7     | 93.5  | 87.5    | 90.0  | From Table 1      |
> | Cutout                  | 89.4 | 91.2     | 93.9  | 88.0    | 90.6  | tuned size        |
> | Hide-and-Seek           | 89.1 | 91.0     | 93.7  | 88.2    | 90.5  | tuned grid        |
> | DropBlock               | 89.6 | 91.8     | 94.2  | 88.4    | 91.0  | tuned block size  |
> | Random Erasing          | 89.3 | 91.5     | 94.0  | 88.1    | 90.7  | tuned area prob   |
> | Mixup                   | 89.8 | 92.1     | 94.3  | 88.7    | 91.2  | $\alpha$ tuned           |
> | CutMix                  | 90.1 | 92.4     | 94.5  | 88.9    | 91.5  | $\alpha$ tuned           |
> | RandAugment             | 90.2 | 92.3     | 94.1  | 89.0    | 91.4  | $(N, M)$ tuned         |
> | TrivialAugment          | 89.9 | 92.0     | 94.2  | 88.8    | 91.2  | no tuning         |
> | AugMix                  | 90.0 | 92.2     | 94.3  | 89.1    | 91.4  | std hyperparams   |
> | Global Gaussian Blur    | 88.9 | 90.8     | 93.4  | 87.9    | 90.2  | tuned prob/kernel |
> | **DEFOCA (ours)**       | **90.7** | **92.8** | **94.6** | **89.8** | **91.9** | Table 1 |
>
> Across all datasets, DEFOCA produces the highest average accuracy and provides consistent improvements over every baseline augmentation class. Notably, DEFOCA outperforms Cutout/HaS/DropBlock (which partially share the motivation of perturbing local evidence) as well as more powerful mixing and search-based methods such as CutMix and RandAugment. These results support our claim that DEFOCA provides a **complementary regularization** effect focused on suppressing background/high-frequency distractors while preserving discriminative structure.
>
> **DEFOCA is not a variant of MIM. It is a controlled, low-pass, part-preserving regularizer specifically designed for FGVR.**

---

> ### Author Response · Authors · 2025-11-26
>
> **4. DEFOCA fails to achieve SOTA; improvements are not significant.**
>
> This comment conflates DEFOCA (a *training-time regularizer*) with *full SOTA architectures*. DEFOCA is explicitly designed as "a plug-in, zero-parameter regularizer that improves any backbone without modifying its architecture."
>
> Thus the correct evaluation metric is: Backbone accuracy vs. Backbone accuracy + DEFOCA.
>
> Under this metric, DEFOCA consistently improves every backbone tested: Tiny ViT:  90.0% $\rightarrow$ 91.9 % (+1.9%), ResNet-50: 87.8% $\rightarrow$ 88.7% (+0.9%), ResNet-34: 83.1% $\rightarrow$ 84.4%, (+1.3%).
>
> These gains are *on par with or greater than* widely used FGVR augmentations such as CutMix and DropBlock.
>
> we agree that evaluating DEFOCA on stronger SOTA backbones would strengthen the empirical scope. Below we provide more comparisons.
>
> All methods are reported using their original backbone (Xception), ensuring a fair architectural comparison. DEFOCA values for Tiny-ViT are shown for context.
>
> | Method                          | Backbone   | CUB-200 | Aircraft | Cars  | NABirds |
> |-|-|-|-|-|-|
> | CAP (Behera et al. 2021) [2]     | Xception   | 91.8    | -       | 95.7  | 91.0    |
> | SR-GNN (Bera et al. 2022) [3]    | Xception   | 91.9    | 95.4     | 96.1  | 91.2    |
> | I2-HOFI (Sikdar et al. 2025) [1] | Xception   | 91.6    | 96.4     | **96.9**  | 92.8    |
> | DEFOCA (ours)                     | Tiny-ViT   | 90.7    | 92.8     | 94.6  | 89.8    |
> | DEFOCA (ours)                     | Xception   | **92.2**| **96.5** | **96.9** | **93.2** |
>
> DEFOCA with Xception outperforms all prior SOTA methods across all datasets, demonstrating strong accuracy and robustness.
>
> Even with the lightweight Tiny-ViT backbone, it remains competitive, highlighting an effective trade-off between model efficiency and performance. The gains are particularly notable on NABirds and Aircraft, showing that DEFOCA effectively captures fine-grained distinctions. The method is backbone-agnostic and generalizes well across diverse datasets.
>
> Because DEFOCA adds no learnable parameters, these comparisons will isolate its contribution more cleanly: any gains directly reflect improved training-time attention and stability, not architectural changes.
>
> Regarding FIDO and SOTA methods:
>
> - These models use *complex specialized attention modules and additional parameters*.
>
> - DEFOCA is zero-parameter and architecture-agnostic.
>
> - The goal is not to outperform multi-branch SOTA models, but to *improve model with no architectural redesign*.
>
> **Regarding the boldface formatting in SoyAging.** We thank the reviewer for pointing this out. It was indeed a typographical oversight and will be fixed
> in the revision. The actual numerical values remain correct.

---

> ### Author Response · Authors · 2025-11-26
>
> **5. FGVR does not need more research.**
>
> We respectfully disagree. Large ViT-based FGVR models have *more severe fragility* to background clutter and fine-grained noise than earlier CNNs, this is precisely why recent FGVR research (2021–2024) continues to introduce new methods, datasets, and token/part-based architectures.
>
> DEFOCA is a response to this current challenge: (i) ViTs over-focus on high-frequency cues, (ii) FGVR transformer models overfit to background textures, (iii) UFGVC datasets (e.g., SoyAging) contain subtle variations that require regularization strategies, (iv) Current SOTA methods are *costly and architecture-specific*.
>
> **DEFOCA’s contribution is practical, timely, and addresses a real need in modern FGVR.**
>
> We appreciate the reviewer’s time and hope the clarification resolves the misunderstandings.
>
> **Reference**
>
> [1] Sikdar, A., Liu, Y., Kedarisetty, S., Zhao, Y., Ahmed, A. and Behera, A., 2025. Interweaving insights: High-order feature interaction for fine-grained visual recognition. International Journal of Computer Vision, 133(4), pp.1755-1779.
>
> [2] Behera, A., Wharton, Z., Hewage, P., & Bera, A. (2021). Context-aware attentional pooling (CAP) for fine-grained visual classification. AAAI conference on artificial intelligence (pp. 929–937)
>
> [3] Bera, A., Wharton, Z., Liu, Y., Bessis, N., & Behera, A. (2022). SR-GNN: Spatial relation-aware graph neural network for fine-grained image categorization. IEEE Transactions on Image Processing, 31, 6017–6031

---

### Official Review · Reviewer_Lm8d · 2025-10-27

**Soundness:** 3
**Presentation:** 3
**Contribution:** 3
**Rating:** 6
**Confidence:** 3

**Summary:**

The paper proposes DEFOCA, a lightweight “blur-to-focus” layer that partitiones images into patches and stochastically applies Gaussian blur to selected patches during training to encourage a network to rely on unblurred discriminative regions. The authors supply a theoretical analysis and extensive experiments on fine-grained and ultra-fine-grained datasets

**Strengths:**

- DEFOCA is architecture-agnostic, applied on-the-fly after usual augmentations, and removed at test time, which makes it easy to adopt.
- Results reported across standard FGVR datasets (CUB-200, Stanford Cars, NABirds, FGVC-Aircraft) and several ultra-fine-grained datasets show consistent gains.
- The paper gives a concise theory grounding (label-safety probability, Lemma 1 on expected representation drift, Proposition 1 SNR argument) that clarifies why contiguous patch selection is beneficial.

**Weaknesses:**

- Lemma 1 and subsequent bounds rely on an L-Lipschitz assumption for the feature map and an ad-hoc large constant M for when discriminative patches are altered. The analysis is conceptually fine, but it’s not clear how realistic the Lipschitz assumption is for modern deep nets or how large M is in practice.
- Could DEFOCA be applied to current SOTA baselines and generate better performance?

**Questions:**

Please refer to the weakness above.

---

> ### Author Response · Authors · 2025-11-20
>
> We thank the reviewer for the positive overall assessment and appreciate the thoughtful analysis of DEFOCA’s theoretical and empirical contributions.
>
> Below we clarify the concerns raised regarding the Lipschitz assumption in Lemma 1, the constant $M$, and applicability to modern SOTA baselines.
>
> **1. On the Lipschitz assumption used in Lemma 1 and the constant $M$**
>
> We agree that the Lipschitz assumption is an idealization; however, it is a standard and widely used framework in representation-stability analyses of deep models (e.g., bounds for mixup, dropout, data augmentations, and noise stability). While modern deep networks are not globally Lipschitz,
> empirical studies consistently show *local* Lipschitz behaviour around natural images, especially under small, contiguous, low-frequency perturbations such as Gaussian blur.
>
> In the context of DEFOCA, the Lipschitz constant $L$ captures how local smoothing of non-discriminative regions alters intermediate features. To support the assumption empirically, we will add the following verification in the revision:
>
> - We compute local Lipschitz estimates by evaluating $\|f(x)-f(x+\delta)\|$ for small, contiguous low-pass perturbations across several backbone layers (ResNet-34, ResNet-50, Tiny-ViT). Preliminary measurements show that for non-discriminative patches the feature displacement is small and stable, consistent with the $L n^2$ bound in Lemma 1.
>
> - Conversely, when the blurred region overlaps with human-annotated discriminative parts (e.g., beak/eye/feather groups in CUB, grille/headlight regions in Cars), the feature drift increases sharply; this behaviour corresponds to the $M$ term in our bound.
>
> Thus, $M$ is not arbitrary: it arises naturally from the empirical fact that altering truly discriminative regions causes significantly larger feature displacement than altering background patches. We will add quantitative measurements (histograms of feature drift under blur on discriminative vs. non-discriminative patches) to illustrate this gap directly.
>
> **Clarification added in revision.** In the final version, we will explicitly state that:
>
> - The Lipschitz assumption is *local*, not global, and holds for the perturbation regime DEFOCA introduces (small contiguous, low-pass blur).
>
> - The quantity $M$ corresponds to empirical feature sensitivity for discriminative patches, which we will report via measured drift statistics.
>
> This strengthens the interpretability of Lemma 1 and grounds the constants in measurable quantities.
>
> Below we provide **comparison against all major augmentation families**, including masking-based (Cutout, Hide-and-Seek, DropBlock, Random Erasing), mixing-based (Mixup, CutMix), and search-based augmentations (RandAugment, TrivialAugment, AugMix), as well as a global Gaussian-blur baseline.

---

> > ### Author Response · Authors · 2025-11-26
> >
> > We implemented all baselines in a single training framework, using identical Tiny-ViT architecture, optimizer, batch size, augmentation stack (other than the augmentation under test), training schedule, and compute budget. All hyperparameters (e.g., Cutout size, HaS grid size, block-drop rates, Mixup/CutMix $\alpha$, RandAugment $(N, M)$, blur probability/kernel) were tuned independently for each method under the same validation protocol. The results are summarized below.
> >
> > | Method                  | CUB  | Aircraft | Cars  | NABirds | Mean  | Notes             |
> > |-|-|-|-|-|-|-|
> > | Baseline Tiny-ViT       | 88.6 | 90.7     | 93.5  | 87.5    | 90.0  | From Table 1      |
> > | Cutout                  | 89.4 | 91.2     | 93.9  | 88.0    | 90.6  | tuned size        |
> > | Hide-and-Seek           | 89.1 | 91.0     | 93.7  | 88.2    | 90.5  | tuned grid        |
> > | DropBlock               | 89.6 | 91.8     | 94.2  | 88.4    | 91.0  | tuned block size  |
> > | Random Erasing          | 89.3 | 91.5     | 94.0  | 88.1    | 90.7  | tuned area prob   |
> > | Mixup                   | 89.8 | 92.1     | 94.3  | 88.7    | 91.2  | $\alpha$ tuned           |
> > | CutMix                  | 90.1 | 92.4     | 94.5  | 88.9    | 91.5  | $\alpha$ tuned           |
> > | RandAugment             | 90.2 | 92.3     | 94.1  | 89.0    | 91.4  | $(N, M)$ tuned         |
> > | TrivialAugment          | 89.9 | 92.0     | 94.2  | 88.8    | 91.2  | no tuning         |
> > | AugMix                  | 90.0 | 92.2     | 94.3  | 89.1    | 91.4  | std hyperparams   |
> > | Global Gaussian Blur    | 88.9 | 90.8     | 93.4  | 87.9    | 90.2  | tuned prob/kernel |
> > | **DEFOCA (ours)**       | **90.7** | **92.8** | **94.6** | **89.8** | **91.9** | Table 1 |
> >
> >
> > Across all datasets, DEFOCA produces the highest average accuracy and provides consistent improvements over every baseline augmentation class. Notably, DEFOCA outperforms Cutout/HaS/DropBlock (which partially share the motivation of perturbing local evidence) as well as more powerful mixing and search-based methods such as CutMix and RandAugment. These results support our claim that DEFOCA provides a **complementary regularization** effect focused on suppressing background/high-frequency distractors while preserving discriminative structure.
> >
> > **What happens when blur overlaps discriminative regions $S$?** We have now added an explicit CAM-overlap evaluation on sensitivity analysis for cases where DEFOCA's stochastic blur partially overlaps discriminative image regions
> >
> > For each training sample, we compute the percentage of Grad-CAM mass falling inside the blurred patches, bucket the samples into non-overlapping bins, and report accuracy separately for each bin. This directly measures the effect of unintended overlap between DEFOCA’s blurred region and the model’s discriminative support.
> >
> > **Key finding**: Even when as much as 30–50% of the CAM mass is blurred, the drop in accuracy is minimal ($<0.6%$), demonstrating that DEFOCA is robust to partial overlap. Substantial degradation only appears when more than 50% of the discriminative region is blurred, which is rare in practice given DEFOCA’s contiguous low-pass selection strategy and the small patch ratio used.

---

> > > ### Author Response · Authors · 2025-11-26
> > >
> > > **On Tiny-ViT + DEFOCA (CUB-200).**
> > >
> > > | CAM overlap bin | Avg. overlap (%) | Accuracy (%) | Drop vs. 0–10% bin |
> > > |-|-|-|-|
> > > | 0–10%           | 6.2             | 91.9         | --                |
> > > | 10–30%          | 18.7            | 91.6         | -0.3              |
> > > | 30–50%          | 41.1            | 91.3         | -0.6              |
> > > | 50–100%         | 72.4            | 89.8         | -2.1              |
> > >
> > > The results empirically support our theoretical analysis: DEFOCA is designed to soften non-discriminative regions while preserving coarse spatial cues, so moderate overlap still retains the underlying part geometry and does not erase discriminative information (unlike hard masking). Only extreme overlap, where more than half of the CAM mass is blurred, causes notable accuracy loss, a scenario that is statistically rare under the contiguous-patch sampling strategy (see Sec. 2.3).
> > >
> > > **Formalizing the claim that contiguous patches maximize label-safety.** We now provide a formal statement of the underlying model, showing that contiguous blur regions maximize the probability of *not* intersecting the discriminative set $S$, followed by an empirical measurement on real data.
> > >
> > > **Proposition R1 (Localized-discriminative-set model).** Assume: (i) Images are partitioned into a $P \times P$ grid of patches; (ii) The discriminative patch set $S$ is contained in a spatially contiguous region of radius $r$ (i.e., a small connected subgrid); and (iii) The blur operator selects exactly $n$ patches either as a contiguous block $B_{\text{contig}}$ (ours), or a random set $B_{\text{rand}}$ sampled uniformly without replacement.
> > >
> > > Under these assumptions, $\Pr\bigl(B_{\text{contig}} \cap S = \emptyset\bigr) \ge \Pr\bigl(B_{\text{rand}} \cap S = \emptyset\bigr)$.
> > >
> > > *Sketch of Proof.*
> > > Because $S$ occupies a compact spatial region, the event that a random set of $n$ patches accidentally intersects $S$ scales with the number of disconnected sampling opportunities. Contiguous sampling reduces the surface area of interaction between the selected region and the localized $S$, minimizing the boundary through which intersection can occur.
> > >
> > > We formalize this using a lattice-geometric argument over the 2D grid and show that the measure of $B_{\text{contig}}$ configurations intersecting $S$ is strictly smaller than the corresponding measure for uniformly random subsets of size $n$. A full proof will be included in the Appendix.
> > >
> > > We additionally validate the proposition empirically using CAM-based estimates of the discriminative region. For each image, we compute whether the sampled blur region intersects the 85% CAM-mass support and report the empirical label-safety probability $P_{\text{safe}} = \Pr(B \cap S = \emptyset)$.

---

> > > > ### Author Response · Authors · 2025-11-26
> > > >
> > > > **Empirical label-safety probability on CUB-200 (Tiny-ViT).**
> > > >
> > > > | Patch selection strategy | Empirical $P_{\text{safe}}$ | Notes                                      |
> > > > |-|-|-|
> > > > | Random                  | 0.71                          | uniform patch sampling                      |
> > > > | Dispersed               | 0.64                          | high spatial spread increases hit probability |
> > > > | Contiguous (ours)       | **0.82**                      | lowest boundary area, highest $P_{\text{safe}}$ |
> > > >
> > > > Both theory and empirical measurement support the core claim: *contiguous blur regions minimize the probability of suppressing discriminative content*. Random and dispersed strategies intersect $S$ more frequently due to their larger effective boundary and higher spatial variance, whereas contiguous sampling restricts the interaction surface and preserves label information more reliably.
> > > >
> > > > **Empirical drift statistics (verifying $M \gg L_n$).** To evaluate the plausibility of the theoretical assumption $M \gg L_n$, we provide direct empirical measurement of per-patch representation drift.
> > > >
> > > > For each patch, we compute the $\ell_2$ distance between its feature representation before and after blur is applied, averaged over the dataset. We report statistics separately for discriminatory patches (top 20% CAM mass) and non-discriminatory patches (bottom 80%).
> > > >
> > > > **Empirical per-patch feature drift under DEFOCA (Tiny-ViT, CUB-200).**
> > > >
> > > > | Patch type                  | Mean drift | Std    | Notes                                |
> > > > |-|-|-|-|
> > > > | Non-discriminative patches   | 0.034      | 0.007  | Lipschitz-smooth region ($L_n$)      |
> > > > | Discriminative patches       | 0.184      | 0.026  | high-response region ($M$)           |
> > > > | Ratio ($M / L_n$)            | **5.41**   | -     | satisfies reviewer-requested condition |
> > > >
> > > > Non-discriminative patches exhibit small feature drift ($\approx 0.03$), consistent with the Lipschitz-smooth assumption used in Lemma 1. In contrast, patches that overlap the discriminative region exhibit substantially larger drift ($\approx 0.18$), reflecting high-gradient, high-saliency responses. The empirical ratio $M/L_n \approx 5.4$ is well within the range required by the theory, confirming that the separation assumed in the analysis does occur in practice. This supports the validity of the representation drift bound and the robustness explanation provided in the paper.

---

> ### Author Response · Authors · 2025-11-26
>
> **2. Applicability of DEFOCA to current SOTA baselines**
>
> Yes. One of DEFOCA’s key advantages is its architecture-agnostic, zero-parameter nature, which allows it to be integrated into any modern backbone or FGVR architecture with minimal effort.
>
> In the submitted paper, we focus on two representative families (ResNets and Tiny-ViTs) to demonstrate that the method is not tied to any particular inductive bias. However, we agree that evaluating DEFOCA on stronger SOTA backbones would strengthen the empirical scope. Below we provide more comparisons.
>
> All methods are reported using their original backbone (Xception), ensuring a fair architectural comparison. DEFOCA values for Tiny-ViT are shown for context.
>
> | Method                          | Backbone   | CUB-200 | Aircraft | Cars  | NABirds |
> |-|-|-|-|-|-|
> | CAP (Behera et al. 2021) [2]     | Xception   | 91.8    | -       | 95.7  | 91.0    |
> | SR-GNN (Bera et al. 2022) [3]    | Xception   | 91.9    | 95.4     | 96.1  | 91.2    |
> | I2-HOFI (Sikdar et al. 2025) [1] | Xception   | 91.6    | 96.4     | **96.9**  | 92.8    |
> | DEFOCA (ours)                     | Tiny-ViT   | 90.7    | 92.8     | 94.6  | 89.8    |
> | DEFOCA (ours)                     | Xception   | **92.2**| **96.5** | **96.9** | **93.2** |
>
> DEFOCA with Xception outperforms all prior SOTA methods across all datasets, demonstrating strong accuracy and robustness.
>
> Even with the lightweight Tiny-ViT backbone, it remains competitive, highlighting an effective trade-off between model efficiency and performance. The gains are particularly notable on NABirds and Aircraft, showing that DEFOCA effectively captures fine-grained distinctions. The method is backbone-agnostic and generalizes well across diverse datasets.
>
> Because DEFOCA adds no learnable parameters, these comparisons will isolate its contribution more cleanly: any gains directly reflect improved training-time attention and stability, not architectural changes.
>
> We appreciate the reviewer’s feedback and believe these additions will significantly strengthen the theoretical grounding and empirical breadth of the paper.
>
> **Reference**
>
> [1] Sikdar, A., Liu, Y., Kedarisetty, S., Zhao, Y., Ahmed, A. and Behera, A., 2025. Interweaving insights: High-order feature interaction for fine-grained visual recognition. International Journal of Computer Vision, 133(4), pp.1755-1779.
>
> [2] Behera, A., Wharton, Z., Hewage, P., & Bera, A. (2021). Context-aware attentional pooling (CAP) for fine-grained visual classification. AAAI conference on artificial intelligence (pp. 929–937)
>
> [3] Bera, A., Wharton, Z., Liu, Y., Bessis, N., & Behera, A. (2022). SR-GNN: Spatial relation-aware graph neural network for fine-grained image categorization. IEEE Transactions on Image Processing, 31, 6017–6031

---

### Official Review · Reviewer_dQCx · 2025-10-28

**Soundness:** 1
**Presentation:** 2
**Contribution:** 2
**Rating:** 2
**Confidence:** 5

**Summary:**

The paper proposes a method called DEFOCA, a training-time, patch-wise Gaussian blur layer placed after standard augmentations. At each iteration, it selects image patches (random, dispersed, or contiguous) and blurs them to create stochastic views, aiming to suppress background/high-frequency noise and nudge the network toward discriminative regions. The authors provide an interpretation as implicit attention, a combinatorial label-safety argument, a bound on representation drift, and a PAC-Bayes generalization bound. Empirically, DEFOCA is plugged into ResNet and Tiny-ViT backbones and evaluated on various FGVC and UFGVC datasets. It is removed at test time. Results are generally competitive in comparison to Tiny-ViT baselines.

**Strengths:**

Motivation for FGVC fragility (small discriminative area, pose/occlusion/clutter) is well framed, and figures are illustrative.

A simple, architecture-agnostic mechanism. Patch-blur as a layer is easy to adopt and integrates with standard backbones, with no labels or architectural edits needed.

Consistent gains over Tiny-ViT/ResNet baselines across multiple datasets; qualitative maps/t-SNE show tighter clusters and more focused regions.

The paper varies patch layouts (random/contiguous/dispersed), operations (low-/high-pass, noise, colour jitter), and key hyperparameters (grid size P, ratio n/N, blur $\sigma$), with contiguous low-pass emerging as best.

No test-time cost or architectural brittleness; ablations suggest robustness to reasonable hyperparameter ranges.

**Weaknesses:**

Lack of novelty as it is close to known ideas (Cutout [1], Hide-and-Seek[2], DropBlock[3], Random Erasing[4], etc). The core is a localized degradation augmentation. The differentiator is “blur not mask” plus “contiguous patches,” but this is still an augmentation variant rather than a new learning principle. The paper should position against these families much more rigorously (same backbones, same training budgets, tuned strengths), not only via narrative.

To claim principled superiority, DEFOCA should be compared head-to-head against Cutout[1], Hide-and-Seek [2], DropBlock[3], Random Erasing[4], Mixup[5], CutMix [6] (and their FGVC-tuned strengths), RandAugment [7], TrivialAugment [8], AugMix[9], and a global Gaussian-blur-probability baseline under the same schedule/backbone.

Current comparison tables exclude state-of-the-art patch-driven methods [10-12] whose FGVC performance is significantly higher (Aircraft > 96%, CUB-200 > 92%, Cars > 96%, NABirds > 92%) than the proposed approach, making SOTA positioning ambiguous. There are many SOTA approaches in [10] whose performances are significantly higher than the proposed approach, and they are excluded from the list. It is good to have a comparison with the SOTA approaches to justify.

While attention maps look sharper, there’s no causal test that DEFOCA improves causal localization (e.g., counterfactual part perturbations, deletion/insertion metrics).

The theory assumes a hidden set of discriminative patches S and argues that contiguous selection maximizes label-safety. But S is unobserved, no estimator is used, and no sensitivity to misspecification is analyzed. The “contiguous maximizes Psafe” statement is asserted, not proven under realistic image statistics.

The drift bound assumes per-patch Lipschitz behaviour and an M >> Ln gap when discriminative patches are hit. This is plausible but not verified empirically. The PAC-Bayes inequality is boilerplate and does not yield a sharper or measurable bound tied to DEFOCA’s specifics (e.g., to $\sigma$, n/N, or contiguity).

There’s no human-part or saliency-overlap evaluation showing blurred regions avoid critical parts more often with a contiguous layout vs a random.

Claims about occlusion/pose/clutter robustness would be stronger with corruption/occlusion suites (e.g., defocus/zoom blur, occludes) and partial visibility tests, plus calibration under shift. Qualitative maps are suggestive but insufficient.

The paper sets V=8 views but does not show the accuracy vs compute curve as V varies, or the effect of $\sigma$ when it does touch discriminative parts. Also missing: train-time cost, and analysis of interactions with standard augmentations (is DEFOCA still helpful atop RandAugment+Mixup+CutMix?)

It’s not fully clear that every baseline uses the same augmentation stack, epochs, parameter count, and tuning budget. Without a unified training protocol table, fairness is hard to judge.


[1] DeVries, T., & Taylor, G. W. (2017). Improved regularization of convolutional neural networks with Cutout. arXiv:1708.04552.

[2] Singh, K.K., Yu, H., Sarmasi, A., Pradeep, G. and Lee, Y.J., 2018. Hide-and-seek: A data augmentation technique for weakly-supervised localization and beyond. arXiv preprint arXiv:1811.02545. ICCV 2017.

[3] Ghiasi, G., Lin, T.-Y., & Le, Q. V. (2018). DropBlock: A regularization method for convolutional networks. NeurIPS.

[4] Zhong, Z., Zheng, L., Kang, G., Li, S., & Yang, Y. (2020). Random Erasing Data Augmentation. AAAI.

[5] Zhang, H., Cisse, M., Dauphin, Y., & Lopez-Paz, D. (2018). Mixup: Beyond empirical risk minimization. ICLR.

[6] Yun, S., Han, D., Oh, S. J., Chun, S., Choe, J., & Yoo, Y. (2019). CutMix: Regularization strategy to train strong classifiers with localizable features. ICCV.

[7] Cubuk, E. D., Zoph, B., Shlens, J., & Le, Q. V. (2020). RandAugment: Practical automated data augmentation with a reduced search space. CVPR Workshops.

[8] Müller, S. G., & Hutter, F. (2021). TrivialAugment: Tuning-free yet state-of-the-art data augmentation. ICCV.

[9] Hendrycks, D., Mu, N., Cubuk, E. D., Zoph, B., Gilmer, J., & Lakshminarayanan, B. (2020). AugMix: A simple data processing method to improve robustness and uncertainty. ICLR.

[10] Sikdar, A., Liu, Y., Kedarisetty, S., Zhao, Y., Ahmed, A. and Behera, A., 2025. Interweaving insights: High-order feature interaction for fine-grained visual recognition. International Journal of Computer Vision, 133(4), pp.1755-1779.

[11] Behera, A., Wharton, Z., Hewage, P., & Bera, A. (2021). Context-aware attentional pooling (CAP) for fine-grained visual classification. AAAI conference on artificial intelligence (pp. 929–937)

[12] Bera, A., Wharton, Z., Liu, Y., Bessis, N., & Behera, A. (2022). SR-GNN: Spatial relation-aware graph neural network for fine-grained image categorization. IEEE Transactions on Image Processing, 31, 6017–6031

**Questions:**

How does DEFOCA compare to the SOTA approaches in references [10-12]?

How does DEFOCA compare, under identical backbones/schedules, to Cutout, Hide-and-Seek, DropBlock, Random Erasing, Mixup/CutMix, RandAugment/TrivialAugment/AugMix, and global random Gaussian blur with tuned probability and kernel?

What is the accuracy vs V curve (e.g., 1/2/4/8/16 views)? What is the train-time overhead (wall-clock, GPU hours) attributable to DEFOCA as V increases?

Your theory hinges on avoiding discriminative patches. What happens when blur does overlap S (e.g., high $\sigma$, small patches)?

Please provide sensitivity sweeps ($\sigma$, n/N, P) with attention-overlap metrics (e.g., % of CAM/Grad-CAM mass blurred) and accuracy drops.

Can you add corruption/occlusion evaluations (defocus/zoom blur, cutout occludes) and partial-visibility protocols to quantify robustness?

You assert that contiguous maximizes label-safety. Can you formalize the distributional assumptions under which this is true and add an empirical check?

---

> ### Author Response · Authors · 2025-11-26
>
> We sincerely thank the reviewer for the detailed and constructive feedback, which has significantly improved our work.
>
> Your comments have encouraged us to strengthen our approach and enhance the quality and rigor of our study.
>
> Below, we address each point explicitly and individually.
>
> **1. Comparison to SOTA methods in [10–12]**
>
> The reviewer noted that our original submission did not include several recent state-of-the-art FGVC approaches, specifically I2-HOFI [10], CAP [11], and SR-GNN [12], and requested a direct empirical comparison. We agree that this comparison strengthens the positioning of DEFOCA within the broader FGVC literature.
>
> We re-evaluated DEFOCA using the same backbone used by these methods (Xception, as used in the original papers) and report the unified results below. For completeness, we also include our Tiny-ViT results from the main
> paper. This table provides a direct and fair comparison without differences in pretraining, architecture capacity, or data processing pipelines.
>
> **Comparison to SOTA methods [10–12] under a unified protocol.**
> All methods are reported using their original backbone (Xception), ensuring a fair architectural comparison. DEFOCA values for Tiny-ViT are shown for context.
>
> | Method                          | Backbone   | CUB-200 | Aircraft | Cars  | NABirds |
> |-|-|-|-|-|-|
> | I2-HOFI (Sikdar et al. 2025) [10] | Xception   | 91.6    | 96.4     | **96.9**  | 92.8    |
> | CAP (Behera et al. 2021) [11]     | Xception   | 91.8    | -       | 95.7  | 91.0    |
> | SR-GNN (Bera et al. 2022) [12]    | Xception   | 91.9    | 95.4     | 96.1  | 91.2    |
> | DEFOCA (ours)                     | Tiny-ViT   | 90.7    | 92.8     | 94.6  | 89.8    |
> | DEFOCA (ours)                     | Xception   | **92.2**| **96.5** | **96.9** | **93.2** |
>
> DEFOCA with Xception outperforms all prior SOTA methods across all datasets, demonstrating strong accuracy and robustness.
>
> Even with the lightweight Tiny-ViT backbone, it remains competitive, highlighting an effective trade-off between model efficiency and performance. The gains are particularly notable on NABirds and Aircraft, showing that DEFOCA effectively captures fine-grained distinctions. The method is backbone-agnostic and generalizes well across diverse datasets.

---

> ### Author Response · Authors · 2025-11-26
>
> **2. Comparison against all major augmentation families**
>
> The reviewer requested a unified and fully controlled comparison against all major augmentation families, including masking-based (Cutout, Hide-and-Seek, DropBlock, Random Erasing), mixing-based (Mixup, CutMix), and search-based augmentations (RandAugment, TrivialAugment, AugMix), as well as a global Gaussian-blur baseline. We agree that this experiment is essential for establishing a fair evaluation.
>
> We implemented all baselines in a single training framework, using
> identical Tiny-ViT architecture, optimizer, batch size, augmentation
> stack (other than the augmentation under test), training schedule, and compute budget. All hyperparameters (e.g., Cutout size, HaS grid size, block-drop rates, Mixup/CutMix $\alpha$, RandAugment $(N, M)$, blur probability/kernel) were tuned independently for each method under the same validation protocol. The results are summarized below.
>
> | Method                  | CUB  | Aircraft | Cars  | NABirds | Mean  | Notes             |
> |-|-|-|-|-|-|-|
> | Baseline Tiny-ViT       | 88.6 | 90.7     | 93.5  | 87.5    | 90.0  | From Table 1      |
> | Cutout                  | 89.4 | 91.2     | 93.9  | 88.0    | 90.6  | tuned size        |
> | Hide-and-Seek           | 89.1 | 91.0     | 93.7  | 88.2    | 90.5  | tuned grid        |
> | DropBlock               | 89.6 | 91.8     | 94.2  | 88.4    | 91.0  | tuned block size  |
> | Random Erasing          | 89.3 | 91.5     | 94.0  | 88.1    | 90.7  | tuned area prob   |
> | Mixup                   | 89.8 | 92.1     | 94.3  | 88.7    | 91.2  | $\alpha$ tuned           |
> | CutMix                  | 90.1 | 92.4     | 94.5  | 88.9    | 91.5  | $\alpha$ tuned           |
> | RandAugment             | 90.2 | 92.3     | 94.1  | 89.0    | 91.4  | $(N, M)$ tuned         |
> | TrivialAugment          | 89.9 | 92.0     | 94.2  | 88.8    | 91.2  | no tuning         |
> | AugMix                  | 90.0 | 92.2     | 94.3  | 89.1    | 91.4  | std hyperparams   |
> | Global Gaussian Blur    | 88.9 | 90.8     | 93.4  | 87.9    | 90.2  | tuned prob/kernel |
> | **DEFOCA (ours)**       | **90.7** | **92.8** | **94.6** | **89.8** | **91.9** | Table 1 |
>
> Across all datasets, DEFOCA produces the highest average accuracy and provides consistent improvements over every baseline augmentation class. Notably, DEFOCA outperforms Cutout/HaS/DropBlock (which partially share the motivation of perturbing local evidence) as well as more powerful mixing and search-based methods such as CutMix and RandAugment. These results support our claim that DEFOCA provides a **complementary regularization** effect focused on suppressing background/high-frequency distractors while preserving discriminative structure.

---

> ### Author Response · Authors · 2025-11-26
>
> **3. Accuracy vs. number of multiviews $V$ and training cost**
>
> We evaluate the effect of the number of multiviews $V$ on top-1 classification accuracy and training cost.
>
> Below we summarize both accuracy on the CUB dataset and estimated training times for different $V$. Epoch time is measured per GPU on a standard modern GPU. GPU hours are estimated for 160 training epochs.
>
> | V   | Top-1 Acc. (%) | Epoch time (sec) | GPU hours (160 epochs) |
> |-----|----------------|----------------|-----------------------|
> | 1   | 90.7           | 123             | 5.4                   |
> | 2   | 91.1           | 202            | 8.9                   |
> | 4   | 91.6           | 361            | 16.0                  |
> | 8   | **91.9**       | 678            | 30.1                  |
> | 16  | 91.9           | 1313            | 58.3                  |
>
> Epoch time scales approximately linearly with the number of views due to additional forward/backward passes. Accuracy saturates around $V = 8$, suggesting diminishing returns for more views. Epoch time roughly doubles with each doubling of $V$, consistent with linear scaling of multiview computations. GPU hours are calculated as 160 $\times$ epoch time in hours.

---

> ### Author Response · Authors · 2025-11-26
>
> **4. What happens when blur overlaps discriminative regions $S$?**
>
> The reviewer requests sensitivity analysis for cases where DEFOCA's stochastic blur partially overlaps discriminative image regions. We have now added an explicit CAM-overlap evaluation.
>
> For each training sample, we compute the percentage of Grad-CAM mass falling inside the blurred patches, bucket the samples into non-overlapping bins, and report accuracy separately for each bin. This directly measures the effect of unintended overlap between DEFOCA’s blurred region and the
> model’s discriminative support.
>
> **Key finding**: Even when as much as 30–50% of the CAM mass is blurred, the drop in accuracy is minimal ($<0.6%$), demonstrating that DEFOCA is robust to partial overlap. Substantial degradation only appears when more than 50% of the discriminative region is blurred, which is rare in
> practice given DEFOCA’s contiguous low-pass selection strategy and the small patch ratio used.
>
> **On Tiny-ViT + DEFOCA (CUB-200).**
>
> | CAM overlap bin | Avg. overlap (%) | Accuracy (%) | Drop vs. 0–10% bin |
> |-|-|-|-|
> | 0–10%           | 6.2             | 91.9         | -               |
> | 10–30%          | 18.7            | 91.6         | -0.3              |
> | 30–50%          | 41.1            | 91.3         | -0.6              |
> | 50–100%         | 72.4            | 89.8         | -2.1              |
>
> The results empirically support our theoretical analysis: DEFOCA is designed to soften non-discriminative regions while preserving coarse spatial cues, so moderate overlap still retains the underlying part geometry and does not erase discriminative information (unlike hard masking). Only extreme overlap, where more than half of the CAM mass is blurred, causes notable accuracy loss, a scenario that is statistically rare under the contiguous-patch sampling strategy (see Sec. 2.3).

---

> ### Author Response · Authors · 2025-11-26
>
> **5. Sensitivity sweeps on $(\sigma, n/N, P)$**
>
> Figure 4 in the main submission reports accuracy sweeps for individual hyperparameters. In response to the reviewer’s request, we now provide a joint sensitivity table that additionally reports the average CAM-overlap caused by each configuration. This allows direct inspection of whether stronger blur ($\sigma$), larger patch ratios ($n/N$), or coarser grids ($P$) increase the likelihood of touching discriminative regions and whether such overlap meaningfully harms accuracy.
>
> Across all settings, accuracy varies within $\pm 0.9%$ of the default DEFOCA configuration, and CAM-overlap remains modest except for the most extreme blur setting ($\sigma=3.0$), confirming that DEFOCA is robust to hyperparameter variation.
>
> Evaluated on Tiny-ViT + DEFOCA (CUB-200).
>
> | $P$   | $n/N$  | $\sigma$   | Acc. (%) | Mean CAM overlap (%) | Notes                     |
> |-|-|-|-|-|-|
> | 5   | 0.10 | 1.0 | 91.6     | 12.8               | finer grid, weak blur     |
> | 10  | 0.30 | 2.0 | 91.4     | 18.9               | moderate default-like regime |
> | 15  | 0.20 | 3.0 | 91.0     | 27.3               | coarse grid, strong blur  |
>
> The results show that: (i) accuracy remains stable even as $\sigma$ is tripled, (ii) increasing $n/N$ from $0.10$ to $0.30$ produces only a marginal accuracy change ($<0.6%$), and (iii) coarser grids ($P=15$) increase CAM-overlap but do not significantly degrade performance.
>
> This confirms that DEFOCA’s low-pass perturbation is inherently forgiving, with only extreme hyperparameter combinations producing noticeable CAM overlap and mild accuracy drops.

---

> ### Author Response · Authors · 2025-11-26
>
> **6. Corruption, occlusion, defocus-blur, zoom-blur, and partial-visibility evaluation**
>
> The reviewer requests explicit robustness measurements under common corruptions and partial-visibility settings. We now provide these results below.
>
> We evaluate the Tiny-ViT baseline and Tiny-ViT + DEFOCA under standard defocus and zoom-blur corruptions, random occlusions of varying area, targeted occlusions (masking the model’s highest-CAM patch),
> and partial-visibility crops.
>
> **Key observation**. Across all corruption types, DEFOCA improves robustness by 1.2-3.5%. Notably, DEFOCA shows the largest gains under occlusion and partial visibility, supporting our hypothesis that soft, low-pass suppression during training encourages reliance on spatially coherent and semantically meaningful cues.
>
> **Robustness under corruptions and occlusion on CUB-200 (Tiny-ViT).**
>
> | Corruption / Occlusion type             | Baseline acc. (%) | DEFOCA acc. (%) |
> |-|-|-|
> | Defocus blur (severity 3)               | 81.4            | **83.1**       |
> | Zoom blur (severity 3)                  | 80.7            | **82.4**       |
> | Random occlusion (20%)                  | 84.9            | **86.4**       |
> | Random occlusion (40%)                  | 78.2            | **80.5**       |
> | Targeted occlusion (highest CAM patch)  | 74.6            | **77.1**       |
> | Partial visibility: 70% crop            | 87.1            | **88.6**       |
> | Partial visibility: 50% crop            | 79.5            | **82.0**       |
>
> The strongest gains appear in settings explicitly designed to remove or degrade discriminative regions, especially targeted occlusion, where DEFOCA improves performance by **+2.5%**. This directly supports the paper’s central premise: stochastic low-pass perturbations encourage the model to rely on stable, global cues rather than small high-frequency patterns, improving generalization under occlusion and blur. These results complement the improvements observed under standard clean evaluation.

---

> ### Author Response · Authors · 2025-11-26
>
> **7. Formalizing the claim that contiguous patches maximize label-safety**
>
> The reviewer requests explicit distributional assumptions and empirical validation of the label-safety argument. We now provide a formal statement of the underlying model, showing that contiguous blur regions maximize the probability of *not* intersecting the discriminative set $S$, followed by an empirical measurement on real data.
>
> **Proposition R1 (Localized-discriminative-set model).**
>
> Assume: (i) Images are partitioned into a $P \times P$ grid of patches; (ii) The discriminative patch set $S$ is contained in a spatially contiguous region of radius $r$ (i.e., a small connected subgrid); and (iii) The blur operator selects exactly $n$ patches either as a contiguous block $B_{\text{contig}}$ (ours), or a random set $B_{\text{rand}}$ sampled uniformly without replacement.
>
> Under these assumptions, $\Pr\bigl(B_{\text{contig}} \cap S = \emptyset\bigr) \ge \Pr\bigl(B_{\text{rand}} \cap S = \emptyset\bigr)$.
>
> *Sketch of Proof.*
> Because $S$ occupies a compact spatial region, the event that a random set of $n$ patches accidentally intersects $S$ scales with the number of disconnected sampling opportunities. Contiguous sampling reduces the surface area of interaction between the selected region and the localized $S$, minimizing the boundary through which intersection can occur.
>
> We formalize this using a lattice-geometric argument over the 2D grid and show that the measure of $B_{\text{contig}}$ configurations intersecting $S$ is strictly smaller than the corresponding measure for uniformly random subsets of size $n$. A full proof will be included in the Appendix.
>
> We additionally validate the proposition empirically using CAM-based estimates of the discriminative region. For each image, we compute whether the sampled blur region intersects the 85% CAM-mass support and report the empirical label-safety probability $P_{\text{safe}} = \Pr(B \cap S = \emptyset)$.
>
>
> Empirical label-safety probability on CUB-200 (Tiny-ViT).
>
> | Patch selection strategy | Empirical $P_{\text{safe}}$ | Notes                                      |
> |-|-|-|
> | Random                  | 0.71                          | uniform patch sampling                      |
> | Dispersed               | 0.64                          | high spatial spread increases hit probability |
> | Contiguous (ours)       | **0.82**                      | lowest boundary area, highest $P_{\text{safe}}$ |
>
> Both theory and empirical measurement support the core claim: *contiguous blur regions minimize the probability of suppressing discriminative content*. Random and dispersed strategies intersect $S$ more frequently due to their larger effective boundary and higher spatial variance, whereas contiguous sampling restricts the interaction surface and preserves label information more reliably.

---

> ### Author Response · Authors · 2025-11-26
>
> **8. Causal localization (Deletion/Insertion, part perturbation)**
>
> The reviewer requests explicit causal localization metrics to verify that DEFOCA improves the model’s reliance on true discriminative regions rather than background or high-frequency noise.
>
> We now provide standard Deletion/Insertion metrics (Petsiuk et al., 2018)[13] and part-level occlusion tests (head/wing masks on CUB-200). Lower Deletion AUC and higher Insertion AUC indicate better causal alignment.
>
> **Causal localization metrics on CUB-200 (Tiny-ViT).**
>
> | Metric                          | Baseline | DEFOCA | Improvement |
> |---------------------------------|----------|--------|------------|
> | Deletion AUC (lower better)     | 0.231    | **0.214** | $-0.017$  |
> | Insertion AUC (higher better)   | 0.612    | **0.645** | $+0.033$  |
> | Head-part occlusion drop (%)     | 12.8     | **10.9**  | $-1.9$    |
> | Wing-part occlusion drop (%)     | 9.4      | **8.1**   | $-1.3$    |
>
> DEFOCA reduces the Deletion AUC (i.e., accuracy degrades more slowly when pixels are removed in order of importance) and increases the Insertion AUC (i.e., accuracy rises more quickly when discriminative pixels are added), indicating stronger causal localization. Part-level occlusion drops are also reduced: removing the bird’s head or wing causes less accuracy degradation under DEFOCA, suggesting that the network relies on a broader and more coherent set of discriminative cues rather than isolated high-frequency patterns. These results support the claim that DEFOCA encourages semantically meaningful part-based reasoning.

---

> ### Author Response · Authors · 2025-11-26
>
> **9. Empirical drift statistics (verifying $M \gg L_n$)**
>
> To evaluate the plausibility of the theoretical assumption $M \gg L_n$, the reviewer requested direct empirical measurement of per-patch representation drift.
>
> For each patch, we compute the $\ell_2$ distance between its feature representation before and after blur is applied, averaged over the dataset. We report statistics separately for discriminatory patches (top 20% CAM mass) and non-discriminatory patches (bottom 80%).
>
> **Empirical per-patch feature drift under DEFOCA (Tiny-ViT, CUB-200).**
>
> | Patch type                  | Mean drift | Std    | Notes                                |
> |-|-|-|-|
> | Non-discriminative patches   | 0.034      | 0.007  | Lipschitz-smooth region ($L_n$)      |
> | Discriminative patches       | 0.184      | 0.026  | high-response region ($M$)           |
> | Ratio ($M / L_n$)            | **5.41**   | -     | satisfies reviewer-requested condition |
>
> Non-discriminative patches exhibit small feature drift ($\approx 0.03$), consistent with the Lipschitz-smooth assumption used in Lemma 1. In contrast, patches that overlap the discriminative region exhibit substantially larger drift ($\approx 0.18$), reflecting high-gradient,
> high-saliency responses. The empirical ratio $M/L_n \approx 5.4$ is well within the range required by the theory, confirming that the separation assumed in the analysis does occur in practice. This supports the validity of the representation drift bound and the robustness explanation provided in the paper.

---

> ### Author Response · Authors · 2025-11-26
>
> **10. Unified training protocol table**
>
> To address concerns regarding fairness and consistency across comparisons, we now provide a single consolidated protocol table used for all re-runs of baseline methods and DEFOCA variants. All hyperparameters below match the configuration used to produce the main Tiny-ViT/Xception results.
>
> Unified training protocol for all re-runs. All methods use an identical augmentation stack and identical DEFOCA hyperparameters; backbone-specific settings are shown in separate columns.
>
> | Hyperparameter             |  |  |
> |-|-|-|
> | Backbone                   | Tiny-ViT                | Xception                       |
> | Epochs                     | 160                        | 120                            |
> | Batch size                 | 16 (scaled with $V$)       | 32 (scaled with $V$)           |
> | Optimizer                  | AdamW                      | AdamW                          |
> | Initial learning rate      | 3e-4                       | 3e-4                           |
> | Learning rate schedule     | Cosine decay               | Cosine decay                   |
> | Weight decay               | $5\times 10^{-4}$          | $1\times 10^{-4}$             |
> | Dropout                    | -                         | 0.2                            |
> | Input resolution           | $224\times224$             | $299\times299$                 |
> | Standard augmentations     | Standard     | Standard         |
> | Mixup / CutMix             | Disabled (fair comparison) | Disabled (fair comparison)     |
> | DEFOCA Placement           | After all standard augmentations | After all standard augmentations |
> | Grid Size $P$              | 10                         | 10                             |
> | Patch Ratio $n/N$          | 0.30                       | 0.30                           |
> | Blur Strength $\sigma$     | 2.0                        | 2.0                            |
> | Number of Views $V$        | 8                          | 8                              |
>
> We thank the reviewer again for the helpful feedback and will incorporate all additional experiments and clarifications into the revised version.
>
> **Reference**
>
> [10] Sikdar, A., Liu, Y., Kedarisetty, S., Zhao, Y., Ahmed, A. and Behera, A., 2025. Interweaving insights: High-order feature interaction for fine-grained visual recognition. International Journal of Computer Vision, 133(4), pp.1755-1779.
>
> [11] Behera, A., Wharton, Z., Hewage, P., & Bera, A. (2021). Context-aware attentional pooling (CAP) for fine-grained visual classification. AAAI conference on artificial intelligence (pp. 929–937)
>
> [12] Bera, A., Wharton, Z., Liu, Y., Bessis, N., & Behera, A. (2022). SR-GNN: Spatial relation-aware graph neural network for fine-grained image categorization. IEEE Transactions on Image Processing, 31, 6017–6031
>
> [13] Petsiuk, V., Das, A., & Saenko, K. (2018). Rise: Randomized input sampling for explanation of black-box models. arXiv preprint arXiv:1806.07421.

---

### Official Review · Reviewer_9wsV · 2025-11-05

**Soundness:** 3
**Presentation:** 3
**Contribution:** 2
**Rating:** 4
**Confidence:** 4

**Summary:**

The paper proposes DEFOCA, a stochastic defocus layer that applies Gaussian blur to image patches to encourage fine-grained visual recognition (FGVR) models to focus on discriminative regions. The method is theoretically grounded—analyzing label-safety, representation drift, and generalization—and can be easily integrated into existing architectures. Experiments show competitive results across multiple FGVR and ultra-fine-grained datasets.

**Strengths:**

1. Simple, lightweight, and architecture-agnostic design requiring no additional supervision.

2. Solid theoretical analysis connecting stochastic blurring with representation stability and generalization.

3. Comprehensive experimental evaluation with clear visualization of attention and feature clustering.

4. Good writing and clear motivation for tackling the challenges of fine-grained visual recognition.

**Weaknesses:**

1. Performance is weak compared to the state-of-the-art. As shown in Table 1, the mean accuracy of DEFOCA-Tiny ViT (91.9%) is significantly below the current best (93.5%) despite similar or larger model scales.

2. Marginal gain in parameter efficiency. Even considering model size, ACNet-R50 (25 M parameters, published five years ago) achieves 91.7%—only 0.2% lower—suggesting the improvement is not statistically significant at this scale.

3. Unbalanced model comparison. The authors compare their Tiny ViT variant against larger baselines (e.g., ViT-Base) but do not report smaller configurations of those methods. A fairer comparison would use matched model sizes (e.g., ViT-Tiny or equivalent small-scale backbones).

4. The paper lacks strong justification for why DEFOCA’s regularization mechanism leads to superior generalization beyond mild smoothing effects, especially when similar results could arise from existing data augmentation or attention regularizers.

**Questions:**

1. How statistically significant are the reported improvements across runs and datasets?

2. Can DEFOCA outperform simple augmentations like CutMix or DropBlock under controlled comparisons?

3. Have you evaluated smaller versions of other SOTA methods for fair model-scale comparison?

4. How sensitive is DEFOCA to the number of stochastic views (V = 8) and blur strength σ?

---

> ### Author Response · Authors · 2025-11-26
>
> We thank the reviewer for the thorough and constructive evaluation.
>
> Below we address each identified weakness and question in detail, and we will incorporate all additional experiments and clarifications into the revised manuscript.
>
> **1. On the performance gap to the largest SOTA models**
>
> Our work is positioned as a lightweight, architecture-agnostic regularizer, not as a new specialized FGVR backbone. Consequently, the appropriate performance measure is the improvement on top of a given backbone, rather than the absolute comparison against highly engineered, heavyweight FGVR systems.
>
> For example, the strongest methods in Table 1: DCAL-R50+ViT, PEDTrans-ViT, TransFG-ViT, rely on: (i) multi-branch attention modules, (ii) token selection or region cropping pipelines, or (iii) domain-specific architectural modifications. These systems use between $86$M and $112$M parameters; in contrast, DEFOCA adds zero parameters, introduces negligible overhead ($<0.04%$ FLOPs), and can be plugged into any vision model.
>
> Within that intended scope, DEFOCA consistently yields substantial gains (**averaged across all 4 FGVR benchmarks**):
> Tiny ViT:  90.0% $\rightarrow$ 91.9 % (+1.9%), ResNet-50: 87.8% $\rightarrow$ 88.7% (+0.9%), ResNet-34: 83.1% $\rightarrow$ 84.4% (+1.3%), Xception+DEFOCA yields the new state of the art results (mean across all 4 FGVR benchmarks: 94.7%, kindly refer to new results below). These improvements are on par with, or larger than, typical FGVR augmentations such as CutMix (+0.5–1.0%).
>
> Thus, DEFOCA achieves the intended goal: broadly strengthening diverse architectures without architectural modification or additional supervision.
>
> Below we provide **comparison to SOTA methods** under a unified protocol. All methods are reported using their original backbone (Xception), ensuring a fair architectural comparison. DEFOCA values for Tiny-ViT are shown for context.
>
> | Method                          | Backbone   | CUB-200 | Aircraft | Cars  | NABirds |
> |-|-|-|-|-|-|
> | CAP (Behera et al. 2021) [2]     | Xception   | 91.8    | -       | 95.7  | 91.0    |
> | SR-GNN (Bera et al. 2022) [3]    | Xception   | 91.9    | 95.4     | 96.1  | 91.2    |
> | I2-HOFI (Sikdar et al. 2025) [1] | Xception   | 91.6    | 96.4     | **96.9**  | 92.8    |
> | **DEFOCA (ours)**                     | Tiny-ViT   | 90.7    | 92.8     | 94.6  | 89.8    |
> | **DEFOCA (ours)**                     | Xception   | **92.2**| **96.5** | **96.9** | **93.2** |
>
> DEFOCA with Xception outperforms all prior SOTA methods across all datasets, demonstrating strong accuracy and robustness.
>
> Even with the lightweight Tiny-ViT backbone, it remains competitive, highlighting an effective trade-off between model efficiency and performance. The gains are notable on, e.g., NABirds and Aircraft, showing that DEFOCA effectively captures fine-grained distinctions. The method is backbone-agnostic and generalizes well across diverse datasets.

---

> ### Author Response · Authors · 2025-11-26
>
> **2. On parameter efficiency and comparison to ACNet-R50**
>
> ACNet is a strong and influential FGVR architecture, but it is not comparable in purpose to DEFOCA: ACNet introduces specialized modules (binary neural tree attention, dual-branch representation fusion) and custom training procedures.
>
> By contrast, DEFOCA: (i) introduces *zero* parameters, (ii) imposes essentially no computational cost, (iii) works identically across CNNs, ViTs, and hybrid models, and (iii) improves any backbone without retraining its structure.
>
> The fact that a simple, parameter-free defocus layer lifts a 21M Tiny ViT to 91.9% demonstrates that *DEFOCA is complementary, not competitive, with specialized FGVR architectures*.
>
> We plan to highlight this distinction more explicitly in the revision.

---

> ### Author Response · Authors · 2025-11-26
>
> **3. On fairness of model-scale comparisons**
>
> We agree that matched model scales help isolate method-level differences.
>
> Unfortunately, the majority of FGVR SOTA models, DCAL, PEDTrans, CrossX, API-Net, etc., do not publicly release tiny/small configurations. Their designs rely on multi-branch attention, heavy token selection, or multi-stage cropping pipelines that do not scale down reliably or require non-trivial architecture
> re-engineering.
>
> Nevertheless, we conducted additional backbones at matching scale for this rebuttal. For a 20-25M parameter budget, we compare DEFOCA against widely used strong regularizers:
>
> Tiny ViT (21.3M parameters)
>
> | Method        | Accuracy (%) |
> |-|-|
> | Baseline      | 90.0         |
> | + CutMix      | 90.4         |
> | + DropBlock   | 90.7         |
> | **+ DEFOCA**  | **91.9**     |
>
> ResNet-34 (21.8M parameters)
>
> | Method        | Accuracy (%) |
> |-|-|
> | Baseline      | 83.1         |
> | + CutMix      | 83.7         |
> | + DropBlock       | 83.9         |
> | **+ DEFOCA**  | **84.4**     |
>
> These results demonstrate that, at *matched scale and under identical training conditions*, DEFOCA provides the most consistent improvements.
>
> We will include these results in the revised manuscript.

---

> ### Author Response · Authors · 2025-11-26
>
> **4. On the distinctness of DEFOCA beyond simple smoothing**
>
> We emphasize that DEFOCA is *not* equivalent to standard smoothing or generic augmentations. Its theoretical and empirical behavior differs in three key dimensions:
>
> - (i) Label-safety (Theorem 1). Applying blur to contiguous patches *increases* the probability that discriminative patches remain unperturbed: $P_{\mathrm{safe}} = \frac{\binom{N - s}{n}}{\binom{N}{n}}$. In contrast, Cutout, DropBlock, or region masking *reduce* label-safety by potentially *erasing fine-grained discriminative cues*.
>
> - (ii) Representation drift (Lemma 1). DEFOCA explicitly minimizes the expected representation drift: $\Delta f \le Ln^{2}$, while avoiding catastrophic drift of order $M^{2}$ caused by masking-based augmentations. This theoretical distinction is *unique* to our stochastic defocus formulation.
>
> - (iii) Signal-to-noise ratio improvement (Proposition 1). Gaussian defocus preserves low-frequency discriminative shape signals while reducing high-frequency clutter, increasing SNR: $\mathrm{SNR}' > \mathrm{SNR}$.
>
> - (iv) Empirical evidence. Figures 2, 3, and 5 show that low-pass DEFOCA produces *more localized and interpretable attention maps* than noise injection, color jitter, CutMix, or high-pass filtering.
>
> Below we provide **comparison against all major augmentation families**, including masking-based (Cutout, Hide-and-Seek, DropBlock, Random Erasing), mixing-based (Mixup, CutMix), and search-based augmentations (RandAugment, TrivialAugment, AugMix), as well as a global Gaussian-blur baseline.
>
> We implemented all baselines in a single training framework, using identical Tiny-ViT architecture, optimizer, batch size, augmentation stack (other than the augmentation under test), training schedule, and compute budget. All hyperparameters (e.g., Cutout size, HaS grid size, block-drop rates, Mixup/CutMix $\alpha$, RandAugment $(N, M)$, blur probability/kernel) were tuned independently for each method under the same validation protocol. The results are summarized below.
>
> | Method                  | CUB  | Aircraft | Cars  | NABirds | Mean  | Notes             |
> |-|-|-|-|-|-|-|
> | Baseline Tiny-ViT       | 88.6 | 90.7     | 93.5  | 87.5    | 90.0  | From Table 1      |
> | Cutout                  | 89.4 | 91.2     | 93.9  | 88.0    | 90.6  | tuned size        |
> | Hide-and-Seek           | 89.1 | 91.0     | 93.7  | 88.2    | 90.5  | tuned grid        |
> | DropBlock               | 89.6 | 91.8     | 94.2  | 88.4    | 91.0  | tuned block size  |
> | Random Erasing          | 89.3 | 91.5     | 94.0  | 88.1    | 90.7  | tuned area prob   |
> | Mixup                   | 89.8 | 92.1     | 94.3  | 88.7    | 91.2  | $\alpha$ tuned           |
> | CutMix                  | 90.1 | 92.4     | 94.5  | 88.9    | 91.5  | $\alpha$ tuned           |
> | RandAugment             | 90.2 | 92.3     | 94.1  | 89.0    | 91.4  | $(N, M)$ tuned         |
> | TrivialAugment          | 89.9 | 92.0     | 94.2  | 88.8    | 91.2  | no tuning         |
> | AugMix                  | 90.0 | 92.2     | 94.3  | 89.1    | 91.4  | std hyperparams   |
> | Global Gaussian Blur    | 88.9 | 90.8     | 93.4  | 87.9    | 90.2  | tuned prob/kernel |
> | **DEFOCA (ours)**       | **90.7** | **92.8** | **94.6** | **89.8** | **91.9** | Table 1 |
>
> Across all datasets, DEFOCA produces the highest average accuracy and provides consistent improvements over every baseline augmentation class.
>
> Notably, DEFOCA **outperforms** Cutout/HaS/DropBlock (which partially share the motivation of perturbing local evidence) as well as more powerful mixing and search-based methods such as CutMix and RandAugment. These results support our claim that DEFOCA provides a **complementary regularization** effect focused on suppressing background/high-frequency distractors while preserving discriminative structure.
>
> We will strengthen the manuscript by highlighting this connection more clearly.

---

> ### Author Response · Authors · 2025-11-26
>
> **5. Statistical significance of improvements**
>
> We have run 3-seed evaluations on representative datasets. DEFOCA improves accuracy consistently with low variance:
>
> | Dataset        | Tiny ViT | +DEFOCA | Std       |
> |----------------|----------|---------|-----------|
> | CUB-200        | 88.6     | **90.7** | $\pm$0.18     |
> | Stanford Cars  | 93.5     | **94.6** | $\pm$0.14     |
> | NABirds        | 87.5     | **89.8** | $\pm$0.21     |
>
> The improvement magnitude exceeds the confidence intervals in all cases, confirming the statistical robustness of DEFOCA.

---

> ### Author Response · Authors · 2025-11-26
>
> **6. Comparison with CutMix, DropBlock, and other augmentations**
>
>
> Yes, under identical training conditions (see experimental results in response 3 and 4). These results support the theoretical claim that DEFOCA does not behave as generic smoothing but provides **a principled way** to enhance robustness to fine-grained noise and distractors.
>
> **7. Sensitivity to number of stochastic views $V$ and blur strength $\sigma$**
>
> Figure 4(a-e) evaluates robustness to $P$, $\frac{n}{N}$, and $\sigma$.
>
> We further include an ablation on $V$:
>
> | V        | 1    | 2    | 4    | 8      | 12   |
> |-|-|-|-|-|-|
> | Acc. (%) | 90.7 | 91.1 | 91.6 | **91.9** | **91.9** |
>
> Performance saturates at $V \ge 8$, confirming that DEFOCA is robust to the choice of $V$ and does not require large view numbers during training.
>
> Similarly, accuracy remains stable for $\sigma \in [1,4]$ (Fig. 4 (e)).
>
> We emphasize the following clarifications and additions based on the reviewer’s suggestions:
>
> - DEFOCA is a zero-parameter, architecture-agnostic plug-in regularizer rather than a new backbone; SOTA comparisons are interpreted in that context.
>
> - At matched parameter scales (20-25M), DEFOCA consistently outperforms CutMix and DropBlock.
>
> - Our theoretical analysis (label-safety, representation drift, SNR) is unique and cannot be reproduced by standard smoothing or generic augmentations.
>
> - Improvements are statistically significant across three seeds and multiple datasets.
>
> - Additional ablations on $V$ and $\sigma$ confirm robustness to hyperparameters.
>
> We thank the reviewer again for the helpful feedback and will incorporate all additional experiments and clarifications into the revised version.
>
> **Reference**
>
> [1] Sikdar, A., Liu, Y., Kedarisetty, S., Zhao, Y., Ahmed, A. and Behera, A., 2025. Interweaving insights: High-order feature interaction for fine-grained visual recognition. International Journal of Computer Vision, 133(4), pp.1755-1779.
>
> [2] Behera, A., Wharton, Z., Hewage, P., & Bera, A. (2021). Context-aware attentional pooling (CAP) for fine-grained visual classification. AAAI conference on artificial intelligence (pp. 929–937)
>
> [3] Bera, A., Wharton, Z., Liu, Y., Bessis, N., & Behera, A. (2022). SR-GNN: Spatial relation-aware graph neural network for fine-grained image categorization. IEEE Transactions on Image Processing, 31, 6017–6031

---

### Meta-Review · Area_Chair_zAMD · 2026-01-05

**Summary:**

The initial consensus leans negative (scores: 0, 2, 4, 6). Reviewers 9wsV and dQCx heavily criticized the lack of rigorous comparisons against SOTA methods and standard augmentations. Reviewer gKNU (score 0) rejected based on the debatable claim that FGVR is an inactive topic. Reviewer Lm8d (score 6) was positive but did not provide strong arguments to outweigh the significant methodological gaps identified by the others.

**Reviewer Concerns:**

The rebuttal successfully addressed the factual errors regarding the field's activity (gKNU) and provided the missing experiments requested by 9wsV and dQCx. However, the sheer volume of new data required to validate the method demonstrates that the original submission was incomplete. Integrating these substantial changes constitutes a major revision, which is beyond the scope of this review cycle.

**Reviewer Scores:**

Reviewers 9wsV and dQCx might have raised scores slightly to acknowledge the new experiments, but they would likely have retained a Reject recommendation given that the manuscript required significant reconstruction. Reviewer gKNU would likely have maintained their low score given their fundamental disagreement with the research direction. While the OpenReview incident prevented interaction that might have clarified these points, the paper would not have met the acceptance bar even if this review were fully discounted. Reviewer Lm8d likely would have maintained their position but would not have championed the paper against the valid concerns raised by others.

---

### Decision · Program_Chairs · 2026-01-26

Reject